# Genomic imprinting in mouse blastocysts is predominantly associated with H3K27me3

Laura Santini[1,11], Florian Halbritter [2,3,11], Fabian Titz-Teixeira [4], Toru Suzuki [5], Maki Asami[5], Xiaoyan Ma[6], Julia Ramesmayer[1], Andreas Lackner [1], Nick Warr[7], Florian Pauler[8], Simon Hippenmeyer [8], Ernest Laue [6], Matthias Farlik [3,9], Christoph Bock [3,10], Andreas Beyer [4], Anthony C. F. Perry [5✉] & Martin Leeb [1✉]

In mammalian genomes, differentially methylated regions (DMRs) and histone marks including trimethylation of histone 3 lysine 27 (H3K27me3) at imprinted genes are asymmetrically inherited to control parentally-biased gene expression. However, neither parent-of-origin-specific transcription nor imprints have been comprehensively mapped at the blastocyst stage of preimplantation development. Here, we address this by integrating transcriptomic and epigenomic approaches in mouse preimplantation embryos. We find that seventy-one genes exhibit previously unreported parent-of-origin-specific expression in blastocysts (nBiX: novel blastocyst-imprinted expressed). Uniparental expression of nBiX genes disappears soon after implantation. Micro-whole-genome bisulfite sequencing (μWGBS) of individual uniparental blastocysts detects 859 DMRs. We further find that 16% of nBiX genes are associated with a DMR, whereas most are associated with parentally-biased H3K27me3, suggesting a role for Polycomb-mediated imprinting in blastocysts. nBiX genes are clustered: five clusters contained at least one published imprinted gene, and five clusters exclusively contained nBiX genes. These data suggest that early development undergoes a complex program of stage-specific imprinting involving different tiers of regulation.

[1] Max Perutz Laboratories Vienna, University of Vienna, Vienna Biocenter, Vienna, Austria. [2] St. Anna Children's Cancer Research Institute (CCRI), Vienna, Austria. [3] CeMM Research Center for Molecular Medicine of the Austrian Academy of Sciences, Vienna, Austria. [4] Cologne Excellence Cluster Cellular Stress Response in Aging-Associated Diseases (CECAD), University of Cologne, Cologne, Germany. [5] Laboratory of Mammalian Molecular Embryology, Department of Biology and Biochemistry, University of Bath, Bath, UK. [6] Department of Biochemistry, University of Cambridge, Cambridge, UK. [7] Mammalian Genetics Unit, MRC Harwell Institute, Harwell, UK. [8] Institute for Science and Technology Austria, Klosterneuburg, Austria. [9] Department of Dermatology, Medical University of Vienna, Vienna, Austria. [10] Institute of Artificial Intelligence and Decision Support, Center for Medical Statistics, Informatics, and Intelligent Systems, Medical University of Vienna, Vienna, Austria. [11] These authors contributed equally: Laura Santini, Florian Halbritter. ✉email: acfp20@bath.ac.uk; martin.leeb@univie.ac.at

For most mammalian genes, both parental alleles are active, but some are expressed from only one allele, determined by its parent-of-origin, and are said to be imprinted. Balanced genome-wide expression of different imprinted genes is critical[1,2] as development stops around the time of implantation in uniparental diploid embryos[1,3,4]. Databases of mouse imprinted genes collectively list 388 genes with parent-of-origin expression bias[1,5–8] (http://www.geneimprint.com/) (Supplementary Dataset 1; see "Materials and Methods" for details). We refer to these as published imprinted genes. Imprinting is associated with chromatin marks that include allele-specific DNA methylation and/or trimethylation of histone H3 lysine 27 (H3K27me3)[9]. DNA methylation-based imprints are associated with differentially methylated regions (DMRs) of the genome. Many DMRs are established during gametogenesis in a *Dnmt3l*-dependent manner to produce germline DMRs (GL-DMRs)[10]. GL-DMRs are key constituents of each of the 24 known imprinting control regions (ICRs) in the mouse[6,11,12].

Although uniparental embryos fail in early development, they form blastocysts from which pluripotent embryonic stem cells (ESCs) can be established. Parthenogenetic haploid (pha) and androgenetic haploid (aha) embryos have been utilised for the derivation of haploid ESCs whose nuclei can be combined with complementary gametes to produce living mice, albeit inefficiently, in semicloning[13–16]. The extent to which poor development in semicloning reflects imprinting instability is unclear. However, it is known that haploid ESCs lose canonical imprints over extended culture periods, which has been leveraged to generate bi-maternal mice[17].

In addition to DNA methylation-based genomic imprinting, a subset of genes with paternal expression bias in mouse preimplantation embryos is maternally enriched for H3K27me3, with no apparent direct dependence on DNA methylation. Most of this H3K27me3-based imprinting is lost in embryonic lineages post-implantation[9,18]. However, the extent of imprinting control by both types of epigenetic mechanism in mouse preimplantation development is unknown. Imprinting defects have severe developmental consequences that can manifest themselves at, or shortly after implantation[19]. It is therefore likely that the imprinting landscape in blastocysts is a critical determinant of normal development, such that dysregulation of blastocyst imprinting has serious detrimental developmental consequences[20].

In this work, we therefore sought to determine parent-of-origin-specific expression in biparental embryos and parent-of-origin-specific DNA methylation in uniparental blastocysts to delineate the imprinting landscape in mouse preimplantation development. Superimposing these and published data on the allele-specific H3K27me3 embryonic landscape reveals the state and provenance of imprinting in preimplantation blastocysts.

## Results

**Assessing parent-of-origin-specific gene expression in F1-hybrid mouse blastocysts**. To delineate parent-of-origin-specific expression bias in mouse blastocysts, we performed allele-specific transcriptome analyses (RNA-seq) of embryonic day 3.5 (E3.5) embryos obtained without in vitro culture from reciprocal *Mus musculus domesticus* C57BL/6 (B6) x *Mus musculus castaneus* (cast) natural mating (Supplementary Fig. 1a). After exclusion of transcripts encoded by the X chromosome, 10,743 robustly expressed transcripts (≥12 reads in at least four out of eight samples) were identified that contained informative single nucleotide polymorphisms (SNPs). The list included 134 of the combined catalogues of published imprinted genes (Supplementary Dataset 1). We further defined the 30 imprinted genes identified in at least three of the

four repositories mentioned above as high confidence (HCon) repository imprints.

One hundred and forty-seven (147) genes exhibited parent-of-origin-specific expression in blastocysts (Fig. 1a and Supplementary Dataset 2; adj. $p \leq 0.1$, DESeq2[21], and further filtered for consistency between the crosses). To increase stringency, we imposed a requirement for a consistent allelic expression ratio of 70:30 or more between parental alleles in at least 60% of embryos for forward and reverse crosses[8,22]. We refer to the first group of 147 genes as blastocyst-skewed expressed (BsX), and the subset that further fulfilled the 70:30 criterion as blastocyst-imprinted expressed (BiX; $n = 105$) genes (Supplementary Fig. 1c).

BsX genes included 36 of the 134 published imprinted genes (henceforth referred to as pubBsX genes; Fig. 1b and Supplementary Dataset 2). Paternal expression was confirmed in independent reciprocal crosses by RT-PCR followed by Sanger sequencing for all eight pubBsX genes tested: *Slc38a4, Peg3, Slc38a1, Jade1, Zfp64, Otx2, Bbx and Epas1* (Fig. 1c; Supplementary Fig. 1e; Supplementary Dataset 3). A large proportion of the published imprinted genes (98 of 134, including *Igf2, H13* and *Commd1*) were absent from the BsX dataset (hereafter referred to as published unconfirmed imprints). We therefore re-evaluated whether these previously reported imprinted genes were indeed expressed equivalently from both alleles, or whether lack of statistical power had excluded them. To this end, we performed a statistical test for equivalent expression from paternal and maternal alleles. Across all analysed genes this identified statistically significant biallelic expression for 5,376 genes (adj. $p \leq 0.1$, $H_0$: absolute log2FC $\geq 1$) (Supplementary Dataset 2), including 24 out of the 134 (18%) published imprinted genes with SNP-containing reads. RT-PCR Sanger sequencing of independent reciprocal cross blastocysts at E3.5 revealed that *Commd1* indeed exhibited mixed expression states (four samples exhibited allele-specific, and two biallelic expression) and *Pon2* was expressed with clear strain bias in blastocysts (Supplementary Fig. 1g, Supplementary Dataset 3). Hence, statistically significant parent-of-origin-specific gene expression was detected in blastocysts for only a quarter of all published imprinted genes, indicating a strong impact of tissue and cell type in defining imprinting patterns during development.

Groups of 71 (56 paternally expressed, 15 maternal) and 111 (78 paternal, 33 maternal) genes that did not include published imprinted genes respectively constitute sets of novel BiX (nBiX) or novel BsX (nBsX) genes (Fig. 1b; Supplementary Fig. 1b, d; Supplementary Dataset 2). RT-PCR Sanger sequencing of independent crosses confirmed uniparental expression of paternally expressed nBiX genes, *Pmaip1, Smyd2, Cblb, Myo1a, Sfxn1* and of the maternally expressed nBiX gene, *Emc2* in E3.5 blastocysts (Fig. 1d and Supplementary Dataset 3). Parent-of-origin-biased expression of tested nBiX genes was lost by E6.5, similar to the recently reported situation for H3K27me3-dependent imprinted genes, *Otx2* and *Bbx*[9], but in contrast to HCon repository imprinted genes, *Slc38a4* and *Peg3*, which maintained uniparental expression at E6.5 (Fig. 1e,f; Supplementary Fig. 1f, Supplementary Dataset 3). We further confirmed parent-of-origin-specific expression of the nBsX genes, *Tmem144* and *Sri* (Supplementary Fig. 1h, i) in independent crosses, suggesting that there was consistent parental-allele expression bias across multiple samples and experiments. Overall, we were able to validate 10 out of 11 tested nBsX genes, including 7 nBiX genes, and 8 out of 8 tested published imprinted genes by RT-PCR Sanger sequencing of independent blastocyst samples (Supplementary Dataset 3). Allele-specific expression analysis of available single-cell RNA-sequencing data confirmed parental expression bias of BiX and BsX genes, despite the signal sparsity (Supplementary Fig. 2a)[18]. In contrast, published unconfirmed

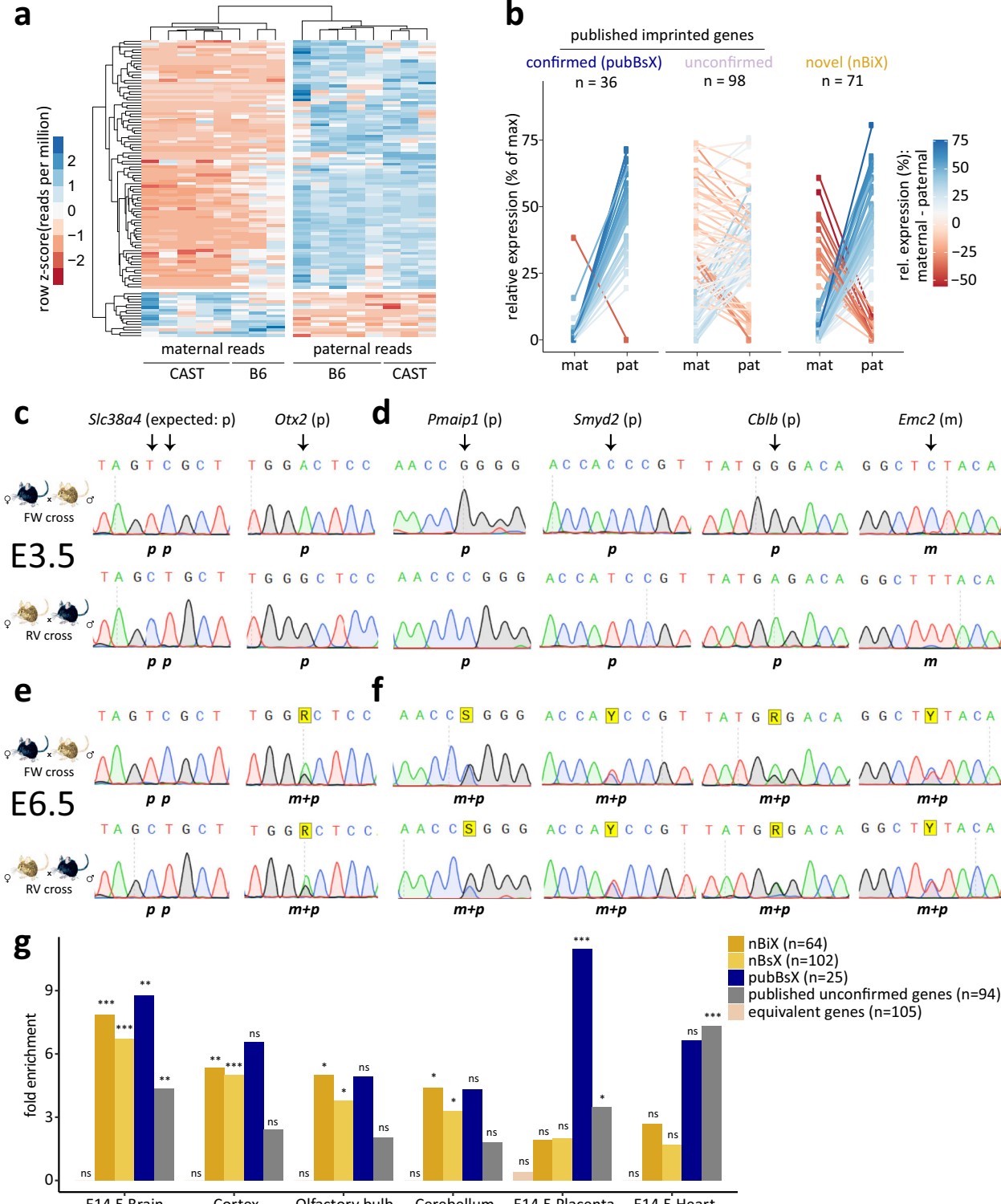

**Fig. 1 Parent-of-origin-specific gene expression in blastocysts. a** Heatmap showing row-normalised expression values of all 105 blastocyst imprinted expressed (BiX) genes. Colour scale indicates Z-scores based on reads per million. Maternal and paternal reads for the same sample are shown in separate columns. **b** Distribution of SNP-containing RNA-seq reads in genetically distinguishable blastocysts on embryonic day 3.5 (E3.5). Comparisons are shown between maternal and paternal alleles in different gene groups [confirmed published imprinted genes (pubBsX), published unconfirmed imprinted genes, novel blastocyst imprint expressed (nBiX)]. Expression values were normalised to the maximum read count per gene and the mean of all replicates is shown. **c**–**f** Electropherogram showing RT-PCR Sanger sequencing-based analysis of allele-specific expression of confirmed published imprinted genes *Slc38a4* and *Otx2* at E3.5 (**c**), of indicated nBiX genes at E3.5 (**d**), of confirmed published imprinted genes *Slc38a4* and *Otx2* at E6.5 (**e**) and of allele-specific expression of indicated nBiX genes at E6.5 (**f**). **g** Barplot showing tissue-specific gene enrichment for different gene groups (nBiX, nBsX, pubBsX, published unconfirmed and equivalently expressed genes), based on analysis with the R package TissueEnrich[64]. FDR-adjusted *p* values were calculated using a hypergeometric test. Only tissues with a significant (adj. *p* < 0.05) enrichment in at least one group of genes are shown. *$p < 0.05$; **$p < 0.01$; ***$p < 0.001$; ns, not significant; *n*, number of genes belonging to each group that are present in the database used for tissue enrichment analysis. Source data are provided as Source Data files.

imprinted genes did not exhibit clear allelic bias in the single-cell data. Our data thus identify sets of nBiX and nBsX genes with high confidence parent-of-origin-specific expression bias.

We next evaluated nBiX and nBsX gene expression by data mining, and in vitro in an ESC model. Both pubBsX and nBiX genes exhibited more variable transcriptional dynamics during the first 24 h of ESC differentiation[23] than other genes that were robustly expressed in blastocysts (Supplementary Fig. 2b). Gene expression data mining revealed an enrichment of transcripts for published imprinted, nBiX and nBsX genes in the developing brain at E14.5. While published imprinted genes exhibited increased expression in placental tissues, no such enrichment was detected for nBiX or nBsX genes (Fig. 1g). The group of genes expressed equivalently from both alleles exhibited neither E14.5 brain nor placental enrichment, indicating the specificity of these results. In sum, we have identified multiple genes with previously unattributed parent-of-origin-specific expression in blastocysts that are enriched in the developing brain.

**Capturing parent-of-origin-specific DNA methylation in uniparental mouse blastocysts.** Imprinted gene expression has been associated with parent-of-origin-specific genomic DNA methylation. To assess whether parentally specified nBiX or nBsX expression could also be explained in this way, we measured genome-wide DNA methylation in individual uniparental parthenogenetic haploid (pha) and androgenetic haploid (aha) E3.5 blastocysts by micro-whole-genome bisulfite sequencing (μWGBS)[15,24,25]. Haploid embryos were selected in an effort to reduce noise that might otherwise have been contributed by different alleles in diploid uniparental embryos. Moreover, uniparental embryos allow unambiguous mapping of μWGBS reads to chromosomes with known parental provenance. Uniparental haploid embryos efficiently formed blastocysts (Supplementary Dataset 4) and contained cells expressing readily detectable OCT4 and CDX2 protein (Supplementary Fig. 3a, b). For comparison, we also derived parthenogenetic haploid ESCs (phaESCs) and androgenetic haploid ESCs (ahaESCs) lines and included three androgenetic, four parthenogenetic and five biparental ESC lines and somatic tissue (kidney) in the μWGBS analysis (Supplementary Fig. 3c). DNA methylation levels could be quantified at 11 to 16.5 million CpGs per sample. Uniparental blastocysts exhibited ~25% global CpG methylation, independently of parental provenance, compared with ~70% CpG methylation in the kidney (Supplementary Fig. 3d) and in line with 20% genomic methylation previously reported for the blastocyst inner cell mass[26]. Investigation of DNA methylation levels unambiguously confirmed parent-of-origin-specific methylation differences at the genomic coordinates of 22 of the 24 known GL-DMRs in haploid uniparental embryos (Fig. 2a). The two GL-DMRs we did not detect across all samples were Snurf/Snrpn, whose DMR lacked coverage in two samples, and Liz1/Zdbf2, whose DMR showed no evidence of DNA methylation in one of the intracytoplasmic sperm injection (ICSI)-derived replicates, confounding unambiguous identification. The uniparental embryo data thus efficiently detected GL-DMRs with a precision that may surpass that obtained for biparental F1-hybrid blastocysts or long read sequencing[26,27] (Supplementary Fig. 3e).

Of six DMRs reportedly acquired in somatic tissues[6], we found that none were uniparentally methylated in blastocysts (Supplementary Fig. 3f), consistent with the acquisition of somatic DMRs post-implantation[28]. However, Nesp, Cdkn1c, Meg3 and Ifg2r promoter-associated somatic DMRs neighboured blastocyst DMRs within 250 kb, and it is possible that these distal DMRs serve to seed methylation of their associated alleles later in development[29].

**DMR erosion in uniparental haploid ESCs.** Four out of five biparental ESC lines maintained GL-DMR methylation levels similar to those in ICSI embryos when cultured in 2i/LIF medium; one female line (ES-f1) exhibited erosion specifically of maternal DMRs (Fig. 2a). We did not detect a strict dependence of imprinting status on cell line, sex or passage number. All biparental lines had strongly reduced DNA methylation of the Gnas ICR. The DNA methylation signal was reduced in some, but not all ESC lines on Slc38a4 and Liz1/Zdbf2 DMRs, suggesting differential stability of DMRs in ESC culture. In contrast, haploid ESC lines cultured in identical conditions typically underwent widespread DMR erosion regardless of their parental provenance. ahaESCs underwent near-complete methylation erosion of paternal H19/Igf2 and Dlk1 GL-DMRs (Fig. 2a). Methylation of the Rasgrf1 GL-DMR was at lower levels than in biparental embryos, indicating ongoing loss of DNA methylation. phaESCs exhibited greater variability in GL-DMR methylation loss than their androgenetic counterparts. In two phaESC lines, most maternal DMRs were maintained at levels similar to those of biparental ESCs, but even then, DMR signals were reduced compared to pha blastocysts, indicating that phaESCs undergo varying levels of DMR loss. Consistent with DMR dysregulation, ahaESCs and phaESCs lacked parent-of-origin-biased expression of the published imprinted genes that we tested (Meg3, Peg3, Slc38a4 and Jade1), although they possessed unperturbed naïve pluripotency and the potential to initiate differentiation (Supplementary Fig. 3g and h). Relative DMR stability in parthenogenotes compared to androgenotes contrasts with previous reports[17], providing support for the idea that imprint erosion does not strictly reflect parental origin. However, imprint erosion in ahaESCs would explain why they failed to support 'semicloning' (Supplementary Dataset 4); embryos produced by injecting ahaESC nuclei into mature oocytes would lack balanced imprinting, resulting in developmental attenuation prior to, or around the time of implantation, as we observed[3,4]. In summary, uniparental haploid ESCs exhibited variable loss of DMRs, even though the DMRs were robustly detectable in uniparental blastocysts.

**Identification of blastocyst-specific DMRs.** Following corroboration of known GL-DMRs, we asked whether our data revealed blastocyst DMRs that had not previously been described. Comparison of genome-wide DNA methylomes from pha and aha blastocysts with those of control, ICSI-derived blastocysts, identified 859 DMRs (dmrseq[30], adj. p ≤ 0.1). Of these, 778 (91%) were maternal (that is, the marks were enriched in parthenogenotes) and 81 (9%) paternal (enriched in androgenotes) (Fig. 2b). The DMRs were associated with 3,664 (7,031) and 392 (779) annotated genes within 100 (or 250) kb windows, respectively (Supplementary Dataset 5); 250 kb is well within the ~300 kb of the Igf2r cluster[2,31]. This unbiased analysis utilising uniparental embryos recovered 23 of the 24 known GL-DMRs. The coordinates of novel DMRs were superimposable upon those of published DMRs (Supplementary Fig. 4a, Supplementary Dataset 5) with the exception of Snurf/Snrpn, where a DMR was detected at a distance of 1 kb from the annotated locus, potentially extending the Snurf/Snrpn GL-DMR (Supplementary Fig. 4b). Only the Liz1/Zdbf2 GL-DMR was not confidently identified in our analysis because it lacked a DNA methylation signal in one of the ICSI samples (Fig. 2a). As was the case for known GL-DMRs, blastocyst DMRs identified here were not maintained in haploid ESCs (Supplementary Fig. 4c and Supplementary Dataset 5).

We next analysed available oocyte and sperm DNA methylome data[26] to determine the developmental origins of the 859 DMRs.

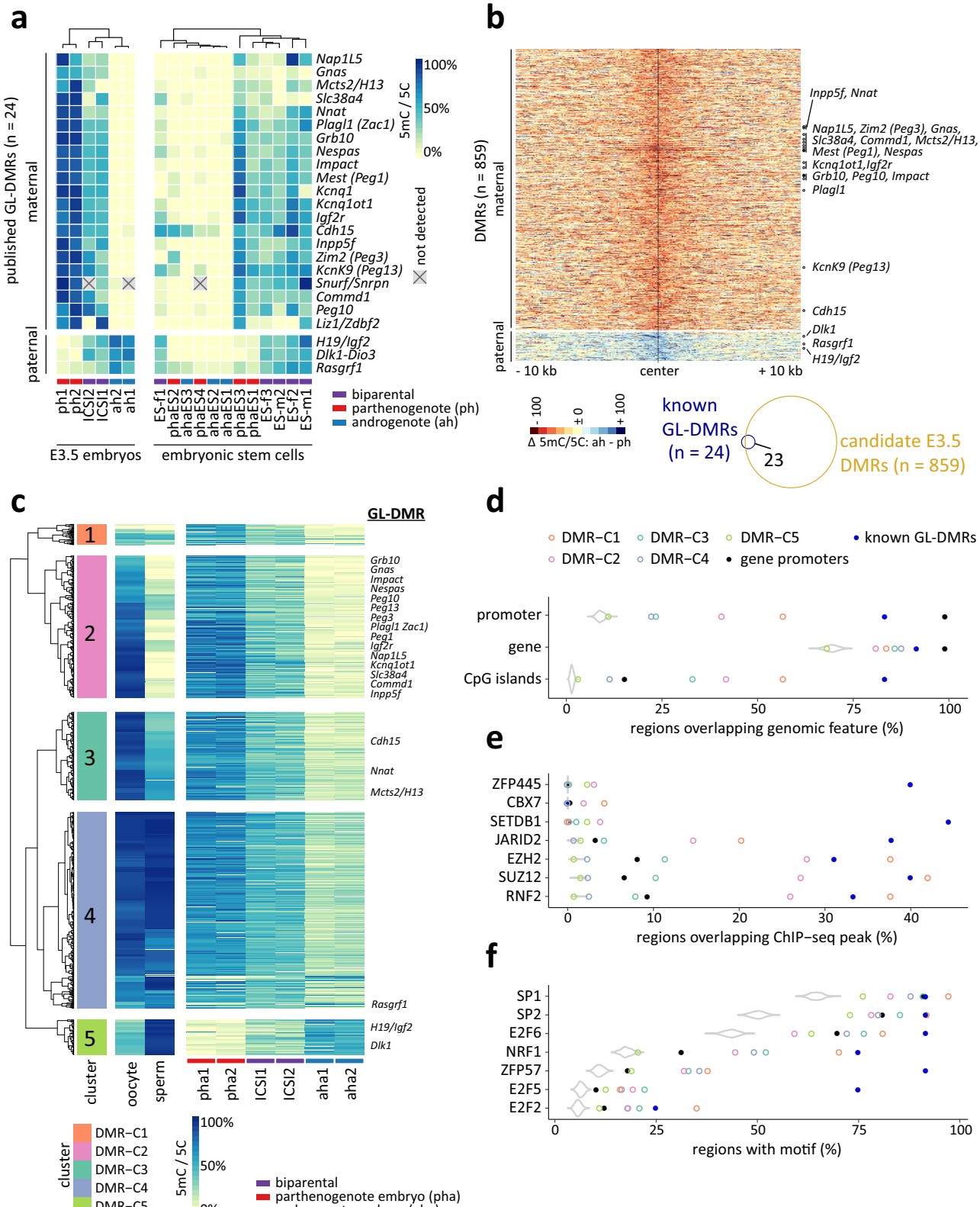

**Fig. 2 Identification of novel DMRs in uniparental embryos. a** Heatmap showing DNA methylation levels for 24 known germline DMRs (GL-DMRs) in blastocyst samples (left) and ESCs. Colour scale represents percentage of 5mC compared to 5C. **b** Heatmap showing DNA methylation signal in a 10 kb window around the centre of all 859 blastocyst DMRs identified in this work (red, maternal DNA methylation; blue, paternal DNA methylation). Known GL-DMRs are indicated rightmost. **c** Heatmap showing DNA methylation levels in all 859 blastocyst DMRs in our blastocyst samples compared to oocyte and sperm DNA methylation from published data[26]. Hierarchical clustering was based on DNA methylation levels in gametes. **d** Distribution of blastocyst DMRs and known GL-DMRs over different genomic features. Gene promoters and 1,000 random sets of regions of comparable size and distribution (from all regions assessed in our DNA methylation analysis, grey) are shown for reference. **e** Locus overlap analysis[32] of published ChIP-seq peaks for blastocyst DMRs and known GL-DMRs. **f** Motif enrichment analysis[37,68] for blastocyst DMRs and known GL-DMRs. Source data are provided as Source Data files.

Of our 778 maternal DMRs, 410 (52%) exhibited oocyte-specific or -biased DNA methylation (DMR clusters C2 and C3, respectively) (Fig. 2c) and 63 of the 81 (76%) paternal DMRs were methylated in sperm genomes (DMR cluster C5). Notably, 349 blastocyst DMRs (41% of the total) were established during preimplantation development by loss of DNA methylation on one parental allele (DMR cluster C4). A minority of blastocyst DMRs (37; 4%) exhibited little or no DNA methylation in oocytes or sperm (DMR cluster C1). These data collectively suggest that most (~60%) differential DNA methylation is encoded within gamete genomes, even though DMRs may become manifest only later in development, mostly by allele-specific reduction of DNA methylation.

Most GL-DMRs (>80%) were located within gene bodies (Fig. 2d) and blastocyst DMRs exhibited a similar distribution. More than 75% of GL-DMRs overlapped with promoters and CpG islands. The overlap with promoters and CpG islands varied for blastocyst DMR clusters (whilst still above background) at 25–60%. We asked which chromatin regulators might interact with the DMRs using two complementary approaches. First, we tested for overlaps between DMRs and binding regions of transcription factors (TFs) and chromatin modifiers by mining 791 published ChIP-seq datasets[32–34]. This identified enrichment on GL-DMRs of the zinc finger protein ZFP445, the H3K9-specific histone-lysine methyltransferase SETDB1 and several components of the Polycomb machinery (Fig. 2e, Supplementary Dataset 5). *Setdb1* establishes H3K9me3, an imprint-associated chromatin mark[35] and *Zfp445* is a primary regulator of genomic imprinting[36]. It has been suggested that Polycomb group proteins help maintain imprints in preimplantation development[9]. Secondly, we searched the DMRs for matches to TF DNA-binding motifs[37]. This detected enrichments for TF cognate binding sequences, including those of *Zfp57*, *E2f5*, and *Nrf1* (Fig. 2f), which were enriched in known GL-DMRs and in all DMR clusters, with the exception of DMR-C5. *Zfp57* synergistically contributes to imprint maintenance with *Zfp445*[36,38] and *Nrf1* has also been implicated in imprinted gene regulation[39]. The strongest enrichments were for TF *Sp1* and *Sp2* motifs for all clusters. *Sp1* DNA binding is proposed to protect CpG islands against DNA methylation by shielding the non-methylated allele from aberrant methylation[40,41]. These findings together suggest shared chromatin regulatory features between DMRs identified here and those of known GL-DMRs.

**Associating DMRs with parent-of-origin-biased expression.** Integrating parent-of-origin-specific blastocyst transcriptome and DNA methylome data have the potential to reveal relationships between imprinted gene expression and DNA methylation. We found that the vast majority of both nBiX and nBsX genes exhibited paternal expression, with maternal DNA methylation at the closest DMR at any distance (Fig. 3a). However, whilst several published imprinted (62/134; 46%) and confirmed published imprinted (13/36; 36%) genes resided close to a DMR (<250 kb), fewer nBiX and nBsX genes were as closely linked (8/71 [11%] and 16/111 [14%], respectively) (Fig. 3b; Supplementary Fig. 5a–c; Supplementary Dataset 5). Moreover, even in published and confirmed published imprint (pubBsX) gene sets, the majority of genes (72/134 [54%] and 23/36 [64%], respectively) were not located near (<250 kb) to a DMR, suggesting that proximal DMRs are not a generically defining feature of imprinted genes in blastocysts.

Large distances between nBiX genes and their nearest DMR (relative to the corresponding distance for published imprinted genes) may reflect long-range tertiary chromatin interactions between DMRs and nBiX loci. We addressed this possibility by

utilising HiC chromosome conformation data from mouse preimplantation embryos[42] and investigated the co-localisation of nBiX or nBsX genes with DMRs in the same topologically associating domain (TAD, Supplementary Fig. 5d). Although this analysis suggested that compartmentalisation of DMRs and BsX genes within TADs in 8 or 64 cell embryos might occur more frequently than expected by chance, the overall number of BsX genes occurring together with a DMR within a TAD was low; our analysis predicted that only 16% of nBiX and 19% nBsX genes were proximated to DMRs in this way (Fig. 3b; Supplementary Dataset 5). These data lead us to conclude that there is little widespread steady-state topological co-compartmentalisation of DMRs and nBiX genes in preimplantation embryos.

**H3K27me3 marks nBsX and nBiX loci.** Parent-of-origin-specific H3K27me3 functions in specifying imprinted gene expression in preimplantation development[9]. We therefore interrogated available data for the inner cell mass (ICM) of mouse embryos at E3.5 to map parent-of-origin-specific H3K27me3 to the transcription start site (TSS) of parent-of-origin-specific genes[43]. We identified parent-of-origin-specific H3K27me3 at the TSS of 741 of the 10,743 genes (7%) whose expression we detected in blastocysts. Forty-seven out of 134 (35%) published imprinted genes showed an enrichment of parent-of-origin-specific H3K27me3 on their respective TSSs, which is less than the overlap with proximal DMRs (46%) (Fig. 3b; Supplementary Dataset 2). This indicates that published imprinted genes are more closely associated with DMRs than with parent-of-origin-specific H3K27me3. However, 23 out of 36 (64%) confirmed published imprinted genes (pubBsX genes) also exhibited parent-of-origin-specific TSS-associated H3K27me3. Thus, H3K27me3 might contribute to regulating parent-of-origin-specific expression of published imprinted genes at the blastocyst stage. The set of published imprinted genes contains previously defined, non-canonically imprinted genes associated with H3K27me3-based silencing[9]. This group of reported non-canonically imprinted genes (hereafter referred to as published non-canonical imprinted genes) exhibited a higher percentage of H3K27me3-decorated TSS (19/29, 65%) compared to the 35% observed for all published imprinted genes. This suggests that allele-specific H3K27me3 association is a general feature of imprinted gene TSSs at the blastocyst stage (Supplementary Fig. 5e).

We therefore extended the H3K27me3 analysis to nBiX and nBsX genes. Both exhibited only low levels of association to DMRs, but they were associated with parent-of-origin-specific H3K27me3 promoter peaks similar to those observed for pubBsX genes (in 54% and 45% of nBiX and nBsX genes, respectively). This implies a senior role for Polycomb (which controls H3K27me3 levels[44]), and a junior one for DNA methylation in regulating (n)BsX gene expression. Five out of 36 (14%) pubBsX genes, and 28 of 71 (39%) nBiX genes were neither associated with parent-of-origin-specific H3K27me3 nor with a proximal DMR.

**H3K27me3 on nBiX and nBsX transcription start sites is encoded in gametes and maternal H3K27me3 is maintained in the epiblast.** To determine whether parent-of-origin-specific H3K27me3 in blastocysts is derived via gametes, akin to GL-DMRs, we investigated published datasets of sperm and oocyte H3K27me3 (Fig. 4a and Supplementary Fig. 6)[43]. Within the maternally expressed group of 15 nBiX and 33 nBsX genes, we observed specific H3K27me3 in sperm in ~25% of cases (Fig. 4b and c). No oocyte-specific H3K27me3 association could be detected for this group of genes. A single confirmed published maternally expressed imprinted gene (*Meg3*) was decorated

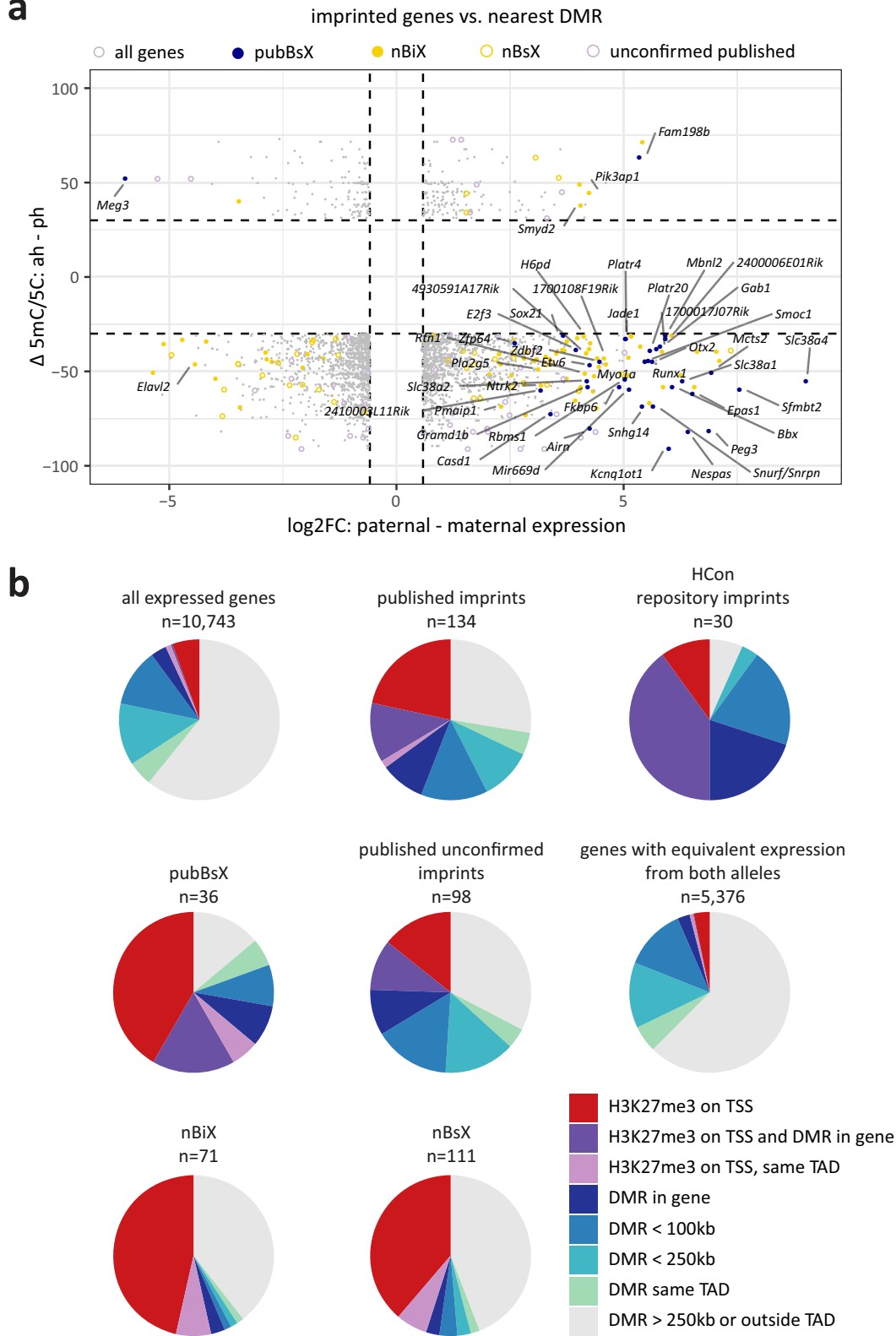

**Fig. 3 Intersecting DMRs and allele-specific H3K27me3 with parental-allele-specific gene expression. a** Comparison of differential DNA methylation in uniparental blastocysts (*y*-axis) and parent-of-origin-specific gene expression (*x*-axis). Published and novel imprinted genes (nBiX and nBsX) are indicated in colour and other genes in grey. Each dot represents one gene associated with its closest DMR. Selected genes are labelled. **b** Pie charts representing all 10,743 genes whose expression was robustly detected, 134 published imprinted genes with expression data, 30 HCon repository imprints, 36 pubBsX genes, 98 published unconfirmed imprinted genes, 5,376 genes that are significantly biallelically expressed in blastocysts, 71 nBiX and 111 nBsX genes. Each chart indicates associations to different genomic features (DMRs and/or parent-of-origin-specific H3K27me3 on TSS). Distances from these genes to their nearest DMR are colour coded. Further colour codes indicate the presence of allele-specific H3K27me3 on the gene promoter (TSS ± 5 kb) or association with a DMR in the same topologically associating domain (TAD), independent of genomic distance. Source data are provided as Source Data files.

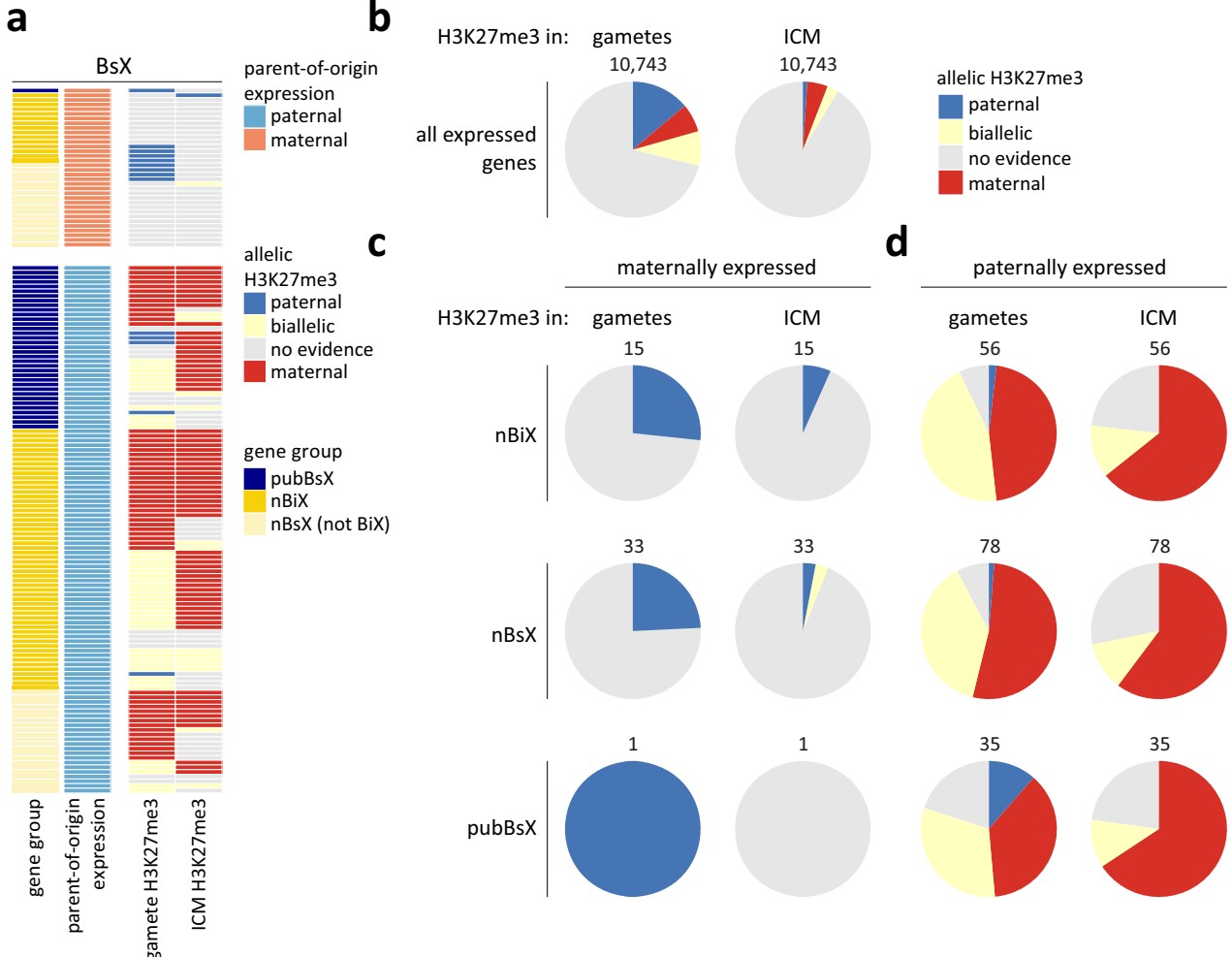

**Fig. 4 Correlation of gamete-specific H3K27me3 with parental-allele-specific gene expression. a** Heatmap showing associations between BsX genes and ICM-allele-specific or gamete-specific H3K27me3. Colour codes distinguish between allelic expression of BsX genes (maternal or paternal), allelic presence of H3K27me3 (on paternal or maternal alleles in the ICM, or in sperm and/or oocyte), and different gene groups (pubBsX, nBiX or nBsX genes). **b** Pie charts illustrating the occurrence of ICM allele-specific or gamete-specific H3K27me3 at the TSS of all 10,743 robustly detected transcripts. **c, d** Pie charts illustrating the occurrence of allele- (in the ICM) or gamete-specific H3K27me3 at the TSS of maternally (**c**) or paternally (**d**) expressed nBiX, nBsX and pubBsX genes. Source data are provided as Source Data files.

by H3K27me3 in sperm. However, paternal allele-specific H3K27me3 was no longer reliably detected at the *Meg3* promoter in blastocysts, indicating either technical limitations of detection or loss of paternal-specific H3K27me3 at the *Meg3* promoter during preimplantation development.

For ~50% of paternally expressed nBiX and nBsX genes, H3K27me3 was localised to the corresponding TSS in oocytes; this percentage further increased for H3K27me3 at maternal alleles in the ICM (Fig. 4b and d). Also, a large proportion (37%) of paternally expressed pubBsX genes were decorated with maternal H3K27me3. After excluding the 20 previously identified, non-canonically imprinted genes[9] from the pubBsX group, four of the remaining 15 genes (27%) exhibited oocyte-specific H3K27me3. Most nBiX or nBsX genes (respectively 45% and 38%) that were paternally expressed at the blastocyst stage but for which we could not identify sperm- or oocyte-specific H3K27me3, nevertheless exhibited H3K27me3 enrichment in both gametes (Fig. 4d). Together, these findings indicate that H3K27me3 is the predominant heritable epigenetic mark responsible for imprinted gene expression in blastocysts.

**Functional dependence of novel and published imprinted gene expression on maternal H3K27me3 and DNA methylation.** Association with a parentally specified epigenetic mark does not necessarily imply a causal relationship to imprinted expression of its corresponding gene. In an effort to evaluate whether associations between epigenetic marks and imprinted gene expression were causal, we harnessed available datasets mapping allele-specific expression in mouse morulae carrying a maternal deletion of either *Dnmt3l* (*mDnmt3l* KO) or *Eed* (*mEed* KO)[45,46]. We reasoned that genes whose allelically skewed expression in control embryos was reduced upon *mDnmt3l* or *mEed* deletion would be regulated by DNA methylation or H3K27me3, respectively. Therefore, we investigated the response of pubBsX and nBsX genes to *mDnmt3l* or *mEed* deletion in preimplantation embryos[45,46] (Fig. 5a and Supplementary Fig. 7a).

Three of the 20 pubBsX genes for which we detected allele-specific expression in control wild-type (WT) embryos exhibited significantly reduced allelic bias in *mDnmt3l* KO embryos (Fig. 5a and b). Allele-specific expression of eight pubBsX genes required *mEed* activity, and a further three were dependent on both *mEed* and *mDnmt3l*. We then separated the group of pubBsX genes

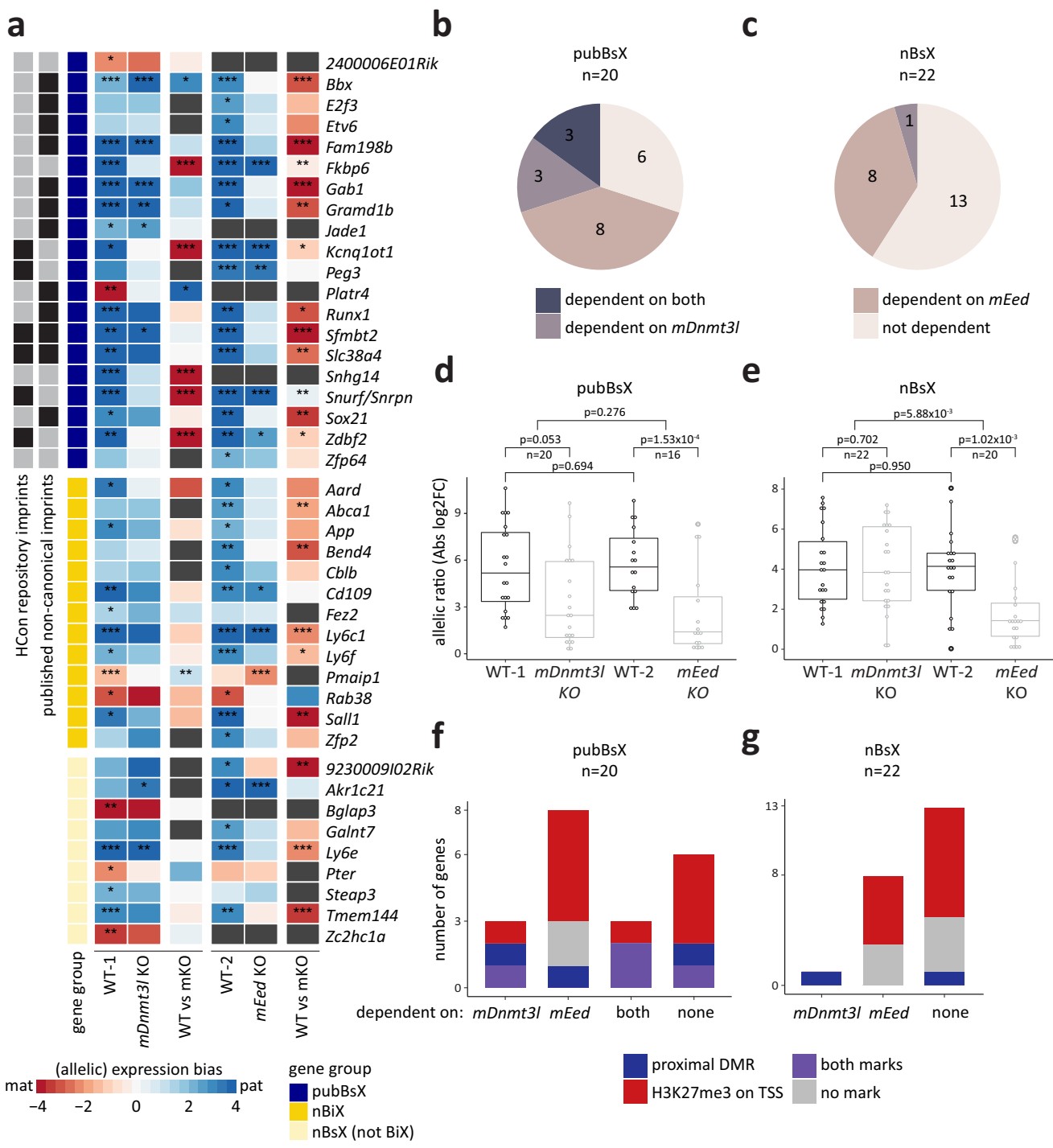

group into published non-canonical imprinted genes[9] and HCon repository imprinted genes (which show a higher level of DMR association, Fig. 3b). As expected, allele-specific expression of a large majority of published non-canonical imprinted genes exhibited *mEed* dependence (Supplementary Fig. 7b). Of the HCon repository genes, imprinted expression of three depended on *mDnmt3l* and two on *mEed*; two genes were dependent on both maternal H3K27me3 and maternal DMRs. The effect of maternal *Eed* KO was exclusively detected in paternally expressed genes.

Of the 22 nBsX genes for which we detected clear allele-specific expression in WT embryos, only the *Pmaip1* gene exhibited

significant dependence on *mDnmt3l* activity (Fig. 5a and c). In contrast, 8 out of 17 nBsX genes (all paternally expressed) were *mEed*-dependent. We also detected a significantly stronger impact of *mEed* KO (compared to *mDnmt3l* KO) on the overall amplitude of parent-of-origin-specific expression of published imprinted and nBsX genes (Fig. 5d, e and Supplementary Fig. 7d). However, the inferred relative contributions of differential DNA methylation and H3K27me3 in maintaining allele-specific expression varied between different gene sets within the group of published imprinted genes: whilst published non-canonical imprinted gene expression[9] clearly depended on *mEed* but not *mDnmt3l*, HCon repository imprints were conversely

**Fig. 5 Functional dependence of novel candidate genes on maternal H3K27me3 or maternal DNA methylation. a** Heatmap indicating allelic expression bias of BsX genes in wild-type (WT) morulae or morulae carrying maternal genetic deletions of either *Dnmt3l* (*mDnmt3l* KO) or *Eed* (*mEed* KO). Colours distinguish between pubBsX, nBiX and nBsX other than BiX genes. pubBsX genes are further divided into genes belonging to the high confidence (HCon) repository imprints or the published non-canonical imprint category (grey/black squares, black indicates membership to the specified category). Only genes with significant allelic bias (adj. $p < 0.1$, DESeq2[21]) in at least one WT morula were included in the analysis (*, adj. $p < 0.1$; **, adj. $p < 0.01$; ***, adj. $p < 0.001$). Allelic expression bias is shown in the first two columns of each WT-mKO set (colour coded from red to blue). The third column of each WT-mKO pair indicates mKO induced changes in the allelic expression bias (colour coded from red to blue; *, adj. $p < 0.05$; **, adj. $p < 0.01$; ***, adj. $p < 0.001$). **b**, **c** Pie charts indicating gene numbers within respective groups (pubBsX (**b**) and nBsX (**c**)) losing parent-of-origin-specific expression following maternal deletion of either *Dntm3l* (dependent on *mDnmt3l*), *Eed* (dependent on *mEed*) or both (dependent on both) in morulae. Genes not dependent on either are also indicated. **d**, **e** Box plots illustrating how allelic ratio (absolute log2FC) of pubBsX (**d**) or nBsX (**e**) genes is affected by maternal deletion of *Dnmt3l* (*mDnmt3l* KO) or *Eed* (*mEed* KO) at the morula stage. Only genes with significant allelic bias (adj. $p < 0.1$) in at least one WT morula were included. Paired two-tailed Wilcoxon signed rank tests were performed for WT *vs* KO comparisons (WT-1 *vs* *mDnmt3l* KO and WT-2 *vs* *mEed* KO). Two-tailed Wilcoxon rank sum tests were performed to compare the two WT datasets (WT-1 *vs* WT-2) and the WT *vs* KO differences between datasets. *p*-values for individual comparisons are indicated in the Figure. All box plots show the 25th percentile, median and 75th percentile; whiskers indicate minimum and maximum values. **f**, **g** Bar charts indicating associations between functional response to loss of either *mDnmt3l* or *mEed* (as defined in Fig. 5b) with physical proximity to DMRs (within 250 kb or in the same TAD) or the presence of TSS-associated (±5 kb) H3K27me3 for pubBsX (**f**) and nBsX (**g**) genes. Source data are provided as Source Data files.

*mDnmt3l*-dependent and *mEed*-independent (Supplementary Fig. 7c). Together, these relationships further underscore a role for the Polycomb system in the imprinting dynamics of preimplantation embryos.

The dependence of allele-specific gene expression of pubBsX and nBsX on *mEed* and *mDnmt3l* correlated with the presence of H3K27me3 at the TSS or a DMR within 250 kb or in the same TAD, respectively (Fig. 5f and g). The group of six pubBsX genes whose expression was apparently unaffected by *mDnmt3l* or *mEed* depletion were nevertheless marked on one parental allele by H3K27me3, a DMR or both (Fig. 5f). Thus, in these cases, imprinting-associated marks do not result in a detectable dependence on *mEed* or *mDnmt3l*. Genes within the set of HCon repository imprinted genes showed an enrichment for H3K27me3, but often also contained a DMR within 250 kb or in the same TAD, indicative of different imprinting mechanisms working in parallel (Supplementary Fig. 7e). This suggests that even in the group of HCon genes, which functionally depend on *mDnmt3l* rather than on *mEed*, most group members exhibit parent-of-origin-specific H3K27me3 enrichment on the TSS. Published non-canonical imprinted genes[9] were almost exclusively marked by H3K27me3 regardless of whether the associated dependency was on *mEed* or *mDnmt3l*.

The single nBsX gene, *Pmaip1*, whose expression was dependent on *mDnmt3l* was associated with a DMR in the same TAD and most (5 out of 8; 62.5 %) *mEed*-dependent nBsX genes were marked by maternal H3K27me3 (Fig. 5 g). Even *mEed*-independent nBsX genes were enriched for allele-specific H3K27me3, one of which was associated with a DMR. Only four nBsX genes that were neither dependent on *mEed* or *mDnmt3l* were also devoid of a proximal DMR or TSS-associated H3K27me3; these included maternally expressed genes, *Rab38*, *Pter* and *Zc2hc1a*, for which dependence on maternal H3K27me3 would not be expected.

In sum, *mEed* and *mDnmt3l* KO datasets suggest that parent-of-origin-specific expression of many nBsX genes critically depends on maternally deposited H3K27me3 rather than on DMRs, implicating the Polycomb system in preimplantation embryo imprint dynamics.

**Identification of imprinted gene clusters in blastocysts.** Imprinted genes are known to reside in genomic clusters regulated by *cis*-acting imprinting control regions (ICRs)[47–49]. We therefore searched for clusters containing at least two novel (nBiX or nBsX) or published imprinted genes within 250 kb, yielding 32 potential ICR clusters (Fig. 6a, b and Supplementary Fig. 8a, b). Twelve clusters contained at least one published imprinted gene that exhibited uniparental expression in blastocysts (Supplementary

Fig. 8a, clusters #1–12). Eight of these also contained a DMR within at least one of its associated genes, less than 10 kb from the TSS. One additional cluster, encompassing the *Slc38a1* gene, contained a DMR within the same TAD. Strikingly, ten clusters of published imprinted genes (including the *Igf2* cluster) lacked significant expression bias in blastocysts, although they contained a DMR within the gene body in nine of the ten cases (Supplementary Fig. 8b, clusters #13–22). This suggests that differential DNA methylation and parent-of-origin-specific expression are sometimes unlinked, at least at the blastocyst stage. A subset of six published imprinted genes in these clusters (including *Commd1* and *Grb10*) contained both a DMR and parental-allele-specific H3K27me3 on their TSSs, but neither of these epigenetic marks elicited allele-specific gene expression. *Commd1* was reported as a brain-specific imprinted gene[50,51]. Thus, although maintenance of an intact ICR may not mediate imprinted expression in preimplantation development, it may mark later parent-of-origin-specific expression in adult tissues.

Our analyses extended five known imprinting clusters by identifying novel imprinted genes close to published examples (Fig. 6a, clusters #23–27). Although these clusters were devoid of proximal DMRs, they were all associated with parent-of-origin-specific H3K27me3. In addition, five novel clusters were identified, exclusively comprising nBsX genes. Four contained at least two protein-coding genes (Fig. 6b, clusters #28-31), and one contained a protein-coding gene and a non-coding RNA (cluster #32). None of these novel clusters possessed a DMR within 250 kb or in the same TAD, and all but one exhibited allele-specific H3K27me3 within the cluster.

## Discussion

By combining allele-specific transcriptomics and uniparental DNA methylome profiling, we have delineated the imprinting status of blastocyst-stage mouse embryos, identifying 859 parent-of-origin-specific DMRs (including 23 out of 24 known GL-DMRs), and 111 genes with previously undescribed parent-of-origin-specific allelic bias (nBsX genes). Of these, 71 exhibited parent-of-origin-specific expression with an allelic ratio of at least 70:30 (nBiX genes).

Parental expression bias in blastocysts was evident for only 36 of 134 published imprinted genes but we detected statistically indistinguishable expression of both alleles for 24 published imprinted genes. Of these, four possessed a DMR within their gene bodies, showing that differential DNA methylation is not sufficient to guarantee uniparental expression in blastocysts. This suggests that hitherto unappreciated tissue- and stage-specified programmes contribute to the regulation of imprinted gene expression.

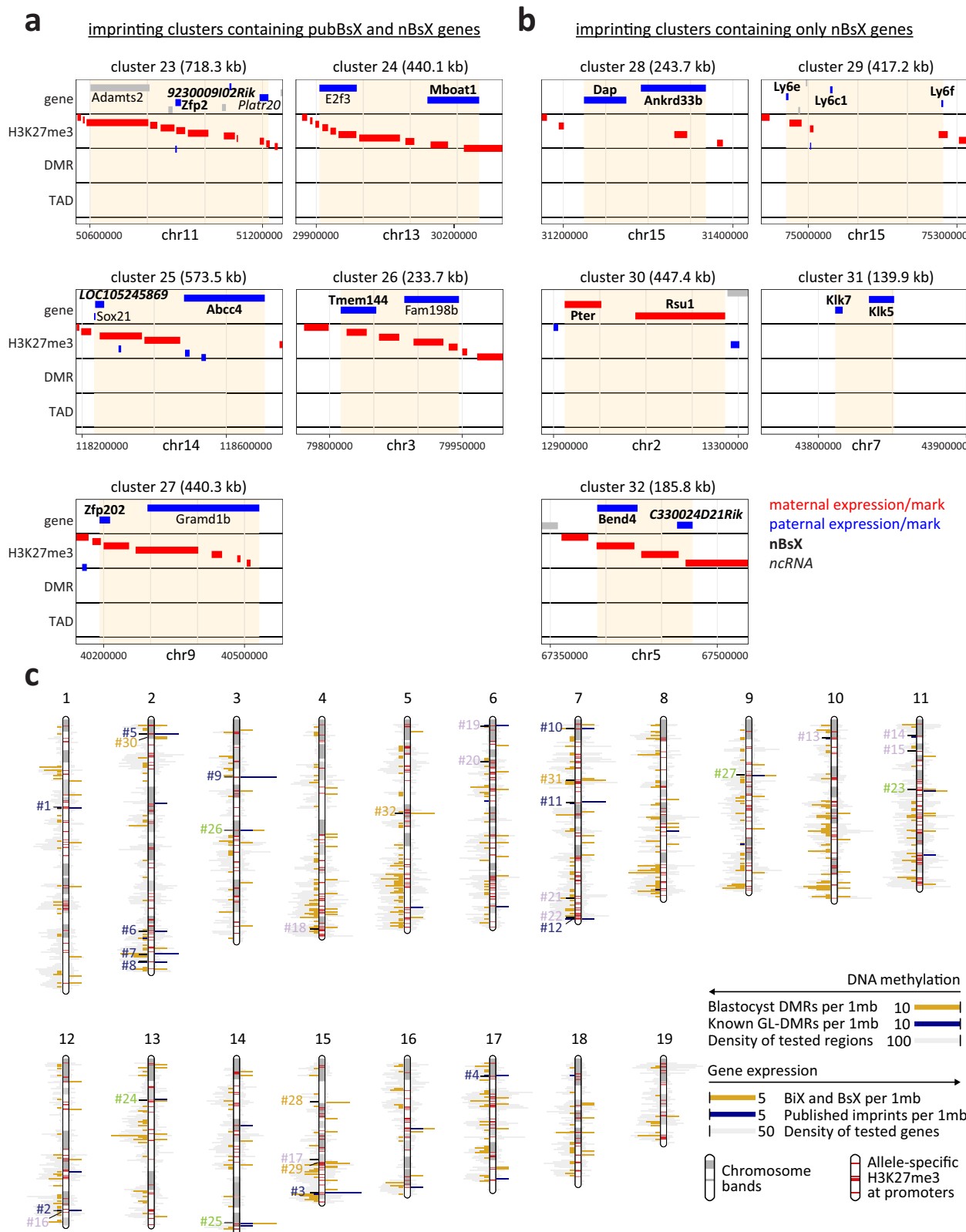

Most (55%) of the DMRs we detected in blastocysts are uniparentally inherited as such and had thus been identified in gamete-specific DNA-methylation analysis. A further 41% of blastocyst DMRs were generated by parent-of-origin-specific reduction of DNA methylation on one (mainly the paternal) parental allele. We also detected gain of allele-specific DNA methylation at some loci that had apparently been unmethylated in gametes. This suggests that parent-of-origin-specific DNA methylation in blastocysts can be encoded in gametes independently of closely linked DNA methylation and later decoded to allow DNA methylation during preimplantation development.

**Fig. 6 Novel imprinting clusters and novel genes in known clusters. a, b** Close-up views of genomic features (genes, DMRs, allele-specific H3K27me3 and allele-specific TADs) for gene clusters containing published imprinted genes containing at least one nBsX gene (clusters 23-27) (**a**) and gene clusters containing only nBsX genes (clusters 28-32) (**b**). Red indicates maternal, and blue paternal allelic expression (genes quadrant; based on our data), maternal/paternal H3K27me3 (H3K27me3 quadrant; based on[43]), maternal/paternal DMR (DMR quadrant; based on our data), maternal/paternal TAD (TAD quadrant; based on[42]). Grey colour for specified genes indicates published imprinted genes for which parent-of-origin-specific expression was not confirmed; grey genes without gene names represent neighbouring genes not included in the cluster analysis. ncRNAs are indicated in italics. nBsX genes are indicated in bold. **c** Visualisation of chromosomal locations of imprinted genes and chromatin marks. Blastocyst DMRs are plotted as bars to the left in gold, known GL-DMRs are shown in blue. The density of tested regions (regions with reads in μWGBS) are plotted in grey. Parent-of-origin expression bias is shown on the right. nBiX and nBsX genes are plotted in gold and published imprinted genes in blue. The density of all robustly expressed genes is plotted in grey. All clusters of (**a**) and (**b**) and Supplementary Figs. 8a and b are indicated [blue, clusters 1–12 (published imprinted genes and at least one pubBsX gene); violet, clusters 13–22 (published imprinted genes with no evidence of parent-of-origin-specific expression in blastocysts); green, clusters 23–27 (published imprinted genes and containing at least one nBsX gene); yellow, clusters 28–32 (containing only nBsX genes)]. The locations of all allele-specific H3K27me3-associated promoters are indicated as red bands overlaid on the chromosome ideograms. Source data are provided as Source Data files.

This work implies that H3K27me3 mediated by Polycomb is the main regulatory mechanism controlling imprinted gene expression in blastocysts. H3K27me3 controls the expression bias of the majority of nBiX and nBsX genes, but also regulates parent-of-origin-specific gene expression of published imprinted genes. Parent-of-origin-specific H3K27me3 is transmitted through the germ line and the majority of nBiX and nBsX genes are decorated by allele-specific H3K27me3. This H3K27me3-dependent imprinting phase appears transient and parent-of-origin-specific expression of all tested nBsX and nBiX genes was lost after implantation.

Our data also show that H3K27me3- and DMR-based imprinting mechanisms regulate partially overlapping gene sets in blastocysts. Indeed, some well-studied imprinted genes for which we could confirm parental-bias in blastocysts, such as, *Snurf/Snrpn*, *Peg3*, *Fkbp6* and *Kcnq1ot1*, harbour both proximal DMRs and H3K27me3 peaks on their TSSs. However, imprinted expression of these genes is not necessarily directly dependent on H3K27me3; for example, while *Kcnq1ot1* and *Fkbp6* are dependent on both *mEed* and *mDnmt3l*, *Snurf/Snrpn* was exclusively dependent on maternal *Dnmt3l*. This suggests that overlapping chromatin profiles of allele-specific DNA methylation and H3K27me3 do not always translate into functional redundancy and that additional tiers of regulation act independently of, or in synergy with, H3K27me3 and/or DMRs to govern imprinted gene expression.

We were able to assign association of virtually all BsX genes with H3K27me3 or DMRs, either functionally, by assessing dependence of imprinted gene expression on *mEed* or *mDnmt3l*, or by the presence of a proximal DMR or H3K27me3 enrichment on the TSS. Only a few cases, mainly concerning maternally expressed genes, lacked evident association with either mechanism. This suggests that Polycomb (specifically, Polycomb Repressive Complex 2, PRC2[52]) and allele-specific DNA methylation machineries cooperate in blastocysts to control imprinted genes expression. The nature of the interaction is unknown. Mechanisms exist to prevent the loss of 5mC from methylated DMRs[53] and functionally analogous pathways might protect H3K27me3 imprints in blastocysts, but which are subsequently de-activated during implantation. We speculate that low global DNA-methylation levels observed at the blastocyst stage[54] could produce less stable imprint regulation by DMRs and that transcriptional repression could be ensured by Polycomb-mediated silencing activity.

Clustering of imprinted genes facilitates coordinated control of parent-of-origin-specific gene expression, such that a given ICR can regulate the expression of multiple genes. We identified five novel imprinted gene clusters and new members of multiple known clusters. All clusters containing nBsX genes lacked blastocyst DMRs detected within 250 kb or present in the same TAD. Six out of the 10 'non-DMR clusters' contained at least one gene associated with H3K27me3, implying that their imprinting is controlled via allele-specific PRC2-mediated histone modification in blastocysts.

We also investigated the maintenance of blastocyst-specific DMRs in vitro in ESCs. Notwithstanding reported imprint instability in 2i culture[55], diploid ESCs stably sustained DMRs with the exceptions of *Gnas* and *Liz1/Zdbf2* loci, at least from 8 to 20 passages in 2i/LIF medium. We also observed a tendency for haploid, but not diploid ESCs to lose DMRs, in contrast to stable imprint maintenance in some human haploid parthenogenetic ESCs[56]. Whether this difference in imprint stability in haploid ESCs reflects differences of species, culture or cell state (e.g. naïve vs primed pluripotent) remains unclear.

In sum, this work provides a detailed compendium containing published and novel imprinted genes and imprinting clusters. It reveals a major contribution of Polycomb-mediated imprinting control in blastocysts, suggesting that imprint regulation in pre-implantation embryos is achieved by both H3K27me3- and DMR-dependent mechanisms. The implication is therefore that there exist different tiers of mechanistically distinguishable, potentially stage-specific imprinting that must be integrated for the healthy development of preimplantation embryos and beyond.

## Methods

**Animals**. Animal procedures, including ethical considerations, complied with the statutes of the Animals (Scientific Procedures) Act, 1986, approved by the University of Bath Animal Welfare and Ethical Review Body and the Biosciences Services Unit. Wild-type mouse strains were bred from stocks in-house or otherwise supplied by Charles River (L'Arbresle, France) or MRC Harwell. The *Mus musculus domesticus* strain B6D2F1 (C57BL/6 [B6] x DBA/2) was generally used as a source of unfertilised metaphase II (mII) oocytes. Some parthenogenotes were produced from *Gt(ROSA) 26Sor*[tm4(ACTB-tdTomato,-EGFP)Luo] (mT) oocytes and we generated a 129/Sv-J line carrying a single, ubiquitously expressed *pCAG-eGFP* transgene (129/Sv-J-eGFP[+/-;57]) and used sperm from hemizygotes to generate androgenetic haploid embryos for ESC derivation. Recipient surrogate mothers were of the strain ICR (CD-1) in embryo transfer. Inter-sub-specific reciprocal crosses were performed by natural mating of B6 *M. m. domesticus* and *M. m. castaneus*.

**Oocytes**. Oocyte collection was essentially performed as described previously[57,58]. Briefly, 8–12-week-old B6D2F1 females were superovulated by standard sequential injection with 5 IU of pregnant mare serum gonadotropin (PMSG) and 5 IU human chorionic gonadotropin (hCG) ~48 h apart. Oocyte-cumulus complexes were collected into M2 medium (Specialty Media, USA) and dispersed with hyaluronidase to denude metaphase II (mII) oocytes, which were washed and cultured in kalium simplex optimised medium (KSOM; Specialty Media, USA) equilibrated in an incubator at 37 °C containing 5% (v/v) humidified $CO_2$ in air until required.

**Sperm**. Preparation of sperm from 129/Sv-J-eGFP[+/−] males was essentially as described previously (Suzuki *et al*., 2014). Epididimides from ~12-week-old males

were minced with fine scissors in nuclear isolation medium (NIM; 125 mM KCl, 2.6 mM NaCl, 7.8 mM $Na_2HPO_4$, 1.4 mM $KH_2PO_4$, 3.0 mM EDTA; pH 7.0) and sperm allowed to disperse. The sperm were washed in NIM and treated in NIM containing 1.0% (w/v) 3-[(3-cholamidopropyl) dimethylammonio]-1-propane-sulfonate (CHAPS) at room temperature. The suspension was gently pelleted and sperm resuspended in ice-cold NIM and held on ice until required. Just prior to ICSI, 50 μl of the sperm suspension was mixed with 20 μl of a solution of 12% (w/v) polyvinylpyrrolidone (PVP, average $M_r \approx 360{,}000$; Sigma, UK).

**Production of uniparental haploid androgenetic embryos.** To establish andro-genic haploid ESC (ahaESC) lines, sperm from 129/Sv-J-eGFP$^{+/-}$ hemizygous males were injected using a piezo-actuated micromanipulator (Prime Tech Ltd., Japan) into B6D2F1 mII oocytes enucleated as described previously[59]: mII oocytes were placed in M2 medium containing 5 μg/ml cytochalasin B and spindles were removed. At least 1 h post-enucleation, sperm heads were injected followed by culture in KSOM for 6 h (37 °C, humidified 5% $CO_2$ [v/v] in air) before recording the morphology of the resultant embryos. Embryos were separated according to whether they possessed a single second polar body (Pb$_2$) and pronucleus (pn) and culture was continued for 3–4 days to be utilised for ahaESC derivation. Parthe-nogenetic embryos were derived by strontium chloride triggered oocyte activation in calcium-free medium followed by in vitro culture to the blastocyst stage in KSOM.

**Production of uniparental haploid parthenogenetic embryos.** Activation of membrane Tomato homozygous ($mT^{+/+}$) transgenic or 129/SvJ oocytes to pro-duce parthenogenetic haploid embryos was by incubation in medium containing 10 mM $SrCl_2$, 16–17.5 h post-hCG, essentially as described[60].

**Sperm microinjection (ICSI).** When required, ~50 μl of sperm suspension was mixed with 20 μl of polyvinylpyrrolidone (PVP, average $M_r \approx 360{,}000$; Sigma-Aldrich) solution (15% [w/v]) and sperm injected (ICSI) into oocytes in a droplet of M2 medium, within ~60 min, essentially as described[58]. Injected oocytes were transferred to KSOM under mineral oil equilibrated in humidified 5% $CO_2$ ([v/v] in air) at 37 °C for embryo culture.

**Establishment and culture of androgenetic and parthenogenetic haploid ES cells.** Haploid ESCs were established and cultured in 2i/LIF medium as previously described[13,14]. Both ahaESCs and phaESCs were recurrently sorted based on DNA content, either by Hoechst staining (15 μg/ml for 15 min @ 37 °C) or based on FCS/SSC parameters[61]. The following ESC lines were used in this study at the passage numbers indicated (except for nucleus injection, below): ES-f1 at p20 (Rex1::GFPd2 reporter cell line); ES-f2 at p17 (129/B6 F1-hybrid female line); ES-m1 at p16 (E14TG2a male ESC line); ES-m2 p8 (male 129 derived ESC line); ES-m3 p8 (male ES cell line of mixed background carrying a floxed, intact Mek1 allele); phaES1 p12 (pha Rex1::GFP reporter ESC line, 129 background[62]); phaES2 p8 (phaESC line 'P1' from a 129 background); phaES3 p8 (phaESC line 'T8', carrying a constitutive tdTomato reporter from a 129 background); phaES4 p12 (phaESC line 'H129-1' from a 129 background[14]); ahaES1 p12 (ahaESC line 'A6GFP' from a 129 back-ground, carrying a constitutively active GFP transgene); ahaES2 p8 (ahaESC line 'A7' from a 129 background); ahaES3 p8 (ahaESC line A11 from a 129 background).

**Androgenetic haploid ESC nucleus injection.** To prepare ahaESCs for nt, semi-confluent cultures at passage number five to seven were sorted utilising FACS. Cell suspensions were mixed with 20 μl of 12% (w/v) PVP solution and injected into mII oocytes essentially as described previously[59]. Following a recovery period of 10–15 min, injected oocytes were activated by incubation at 37 °C under 5% (v/v) humidified $CO_2$ in air for 2–4 h in calcium-free CZB-G medium supplemented with 10 mM $SrCl_2$[57]. After 6~8 h, the number of Pb$_2$ and pn in embryos was determined and those with a single Pb$_2$ and two pn (Pb$_2$-pn2) were placed in a separate drop and culture continued in KSOM. Where appropriate, embryos at the two-cell stage were transferred to pseudopregnant CD-1 (ICR) females[57]. As a proof-of-principle, we generated a cloned offspring by cumulus cell nuclear transfer (nt) essentially as described previously[59].

**Preparation of ahaES cells for nt.** Following culture of FACS-purified ahaES cells at 37 °C in humidified 5% (v/v) $CO_2$ in air, cell suspensions were prepared as previously described[13,61]. Briefly, cells were washed with DMEM medium followed by calcium-free PBS and incubated with trypsin/EDTA for 3 min at 37 °C. Tryp-sinization was quenched by the addition of 5 ml ES/DMEM (DMEM supplemented with 5% [v/v] FCS/LIF[13]) and cells dissociated by gentle pipetting. Single-cell suspensions were pelleted by centrifugation (221g, 5 min) and resuspended in fresh ES/DMEM medium. Single-cell ahaES cell suspensions were placed on ice and used immediately for micromanipulation. In some cases, haploid cells were enriched by FACS sorting immediately prior to micromanipulation. Cell aggregates were removed by passing suspensions though a 50 μm cell strainer (Falcon) into a polypropylene FACS tube (BD). To avoid Hoechst toxicity, we employed SSC and FSC as FACS Aria parameters for haploid and diploid population separation.

Enriched haploid ES cells were collected into an ice-cold FACS tube containing 1 ml ES/DMEM supplemented with serum and immediately used for micro-manipulation. G1 cell selection was further attempted by selecting smaller cells as nucleus donors.

**Immunocytochemistry.** Embryos were fixed with 4% (w/v) paraformaldehyde (Santa Cruz, USA, cas 30525-89-4) and processed using standard immunocytochemistry[60]. Briefly, samples were incubated overnight at 4 °C with anti-Oct4 (1:100 [v/v]; Santa Cruz, USA, H134) or -Cdx2 (1:100 [v/v]; BioGenex Laboratories, USA, MU392A-UC) primary antibodies. Primary antibody incubation was followed by incubation for 1 h at 37 °C with secondary antibody (1:250 [v/v]; Life Technologies Ltd., UK) conjugated to Alexa 488 and/or Alexa 594. Fluorescence of fixed samples was visualised on an Eclipse E600 (Nikon, Japan) microscope equipped with a Radiance 2100 laser scanning confocal system (BioRad, USA), and images processed using Image J (http://imagej.nih.gov/ij/).

**Differentiation assay.** To evaluate differentiation potential of parthenogenetic and androgenetic cells, the expression level of naïve pluripotency and early differ-entiation markers was analysed in comparison to biparental control by RT-qPCR. ES-m1, ES-m2, ES-f1, phaES1, phaES3 phaES4, ahaES1 and ahaES2 cell lines were plated in N2B27 based 2i/LIF medium at a final density of $10^4$ cells/cm$^2$. On the next day, cells were washed with PBS and medium was exchanged to either N2B27 without 2i/LIF to induce differentiation, or fresh N2B27 + 2i/LIF for the undif-ferentiated controls. After 24 h, cells were washed twice with PBS and harvested in RNA Lysis buffer containing 1% (v/v) 2-merchaptothanol and stored at −80 °C before isolation of RNA using the EXTRACTME Total RNA Kit (Blirt). RNA was reverse-transcribed into cDNA using the SensiFAST cDNA Synthesis Kit (Bioline). Expression of pluripotency and early differentiation markers as well as selected published imprinted genes were determined by qPCR using the Sensifast SYBR No Rox-Kit (Bioline). Data analysis was performed using Microsoft Excel (Office 365) and Graphpad Prism (v5.03). Primers are listed in Supplementary Table 1.

**Imprinted gene assignment from RNA-seq data.** Single blastocysts from natural mating reciprocal crosses between M. m. domesticus (B6) and M. m. castaneus (cast) were lysed and RNA extracted. Samples were processed using a SMART-Seq2 compatible protocol as described[24]. RNA-seq reads were aligned to the mm10 genome using the STAR aligner[63] (version2.6.0c). Reads mapping to multiple locations were excluded from further analysis. Only reads covering annotated SNPs for B6 and cast were used for the analysis. The number of reads per gene that could be assigned to one of the strains by the SNP information was taken from an intermediate result of the Allelome.Pro (v1.0) pipeline[8,22] and used to create a count table for all samples from forward (B6 x cast) and reverse crosses (cast x B6). To ensure data quality, samples 4 (which contained fewer than 5% of reads mapping to the reference genome) and 6 (which gave too few total reads) were removed from the analysis pipeline. Moreover, genes on the X chromosome (analysed embryos were not matched for sex) and genes with fewer than ten SNP spanning reads in at least one sample were removed from further analysis. To further exclude the possibility of erroneously calling imprinted genes due to reads assigned to overlapping transcripts, we only included genes if reads could be unambiguously assigned to a specific transcript. We then employed DESeq2 to test for significant differences in maternal and paternal expression (FDR-adjusted p-value ≤ 0.1; H$_0$: log2FC = 0). We further excluded genes that were not robustly expressed. To achieve this, we quantified the reads (including reads without SNPs) overlapping disjoint exons (exons uniquely attributable to one transcript) and removed all transcripts that did not have at least 12 reads in at least four (out of eight) samples. This resulted in a set of 10,743 genes with robust expression. Genes with a |log2(maternal/paternal reads)| > 0.5 that fulfilled previous criteria and were consistently parentally biased across all samples were defined as blastocyst-skewed expressed genes (BsX genes). Genes were considered to be consistently parentally biased if forward and reverse crosses exhibited bias in the same direction (e.g. both directions were biased towards either paternal or maternal) and both directions of the crosses had at least 20 SNP spanning reads assigned between the corresponding samples (reject = 0, Supplementary Dataset 2). Additionally, genes with a 70:30 expression ratio (or greater) in at least 60% of samples in each cross between the parental alleles were considered as BiX genes (blastocyst-imprinted expressed genes). An additional test was performed to identify genes that exhibited significant biallelic expression (FDR-adjusted p-value ≤ 0.1; H$_0$:|log2FC| < 1). The catalogue of imprinted genes inferred from this work is presented in Supplementary Dataset 2. The list of published imprinted genes comprises genes previously reported to be imprinted in the literature[1,8] and genes present in four imprinting repositories (Mousebook [https://www.mousebook.org/], Otago [https://www.otago.ac.nz/IGC], Geneimprint [http://www.geneimprint.com/], Wamidex [https://atlas.genetics.kcl.ac.uk])[5–7], sourced in August 2019; Supplementary Dataset 1). From 388 unique imprinted gene names, 238 were also found in our dataset and could be assigned consistent gene symbols. Of these genes, 10 were located on the X chromosome and 51 were not represented by at least 10 SNP-overlapping reads; these genes were excluded from further analysis, while 178 genes remained in the analysis pipeline. An additional three gene names were associated with predicted genes and hence removed from further analysis. The intersection of this list with

the set of 10,743 robustly expressed genes generated the final list of 134 published imprinted genes.

Of the 76 genes defined as non-canonically imprinted by Inoue et al.[9], 48 were included in the intermediate results from the Allelome.Pro pipeline. The remaining 28 genes not included in the Allelome.Pro output shared a total of only six reads (not restricted to SNP spanning reads) assigned to them between all samples and were therefore too lowly expressed to be included in the analysis. Thirty-six out of the detected 48 genes were located on autosomes. After filtering for robust expression, 29 genes were left and referred to as published non-canonical imprints. These published non-canonical imprints are part of the group of 134 published imprinted genes.

**Tissue enrichment analysis.** To analyse tissue-specific gene enrichment for our candidate genes we used the Bioconductor package TissueEnrich (v1.10.10)[64]. Specifically, we used the teEnrichment function selecting the Mouse ENCODE dataset as the RNA-seq reference dataset and specifying to consider all types of tissue-specific genes (Tissue-Enriched, Tissue-Enhanced and Group-Enriched) in the enrichment analysis., All 10,743 genes that were robustly expressed in our allele-specific RNA-Seq dataset were defined as background. We calculated the tissue-specific gene enrichment analysis for the following group of genes: novel imprinted genes (nBiX and nBsX), published confirmed genes (pubBsX) and published unconfirmed genes, and equivalent genes. For the latter group, we selected the top 105 genes with equivalent expression (ranked by adjusted p-value) to avoid any possible bias due to different genes groups sizes. We considered the enrichment significant when adj. p < 0.05.

**Evaluation of imprinted gene sets.** We obtained allele-specific single-cell gene expression data for oocytes and preimplantation embryos[18] from GEO (GSE80810) and used these data to confirm parentally biased allele-specific expression of published, nBiX, nBsX, but not of published unconfirmed imprinted genes throughout preimplantation development. A comparison of absolute log2FCs between ESCs cultured in 2i and 24 h after induction of differentiation by 2i withdrawal[23] for different groups of genes was utilised to determine whether they exhibited dynamic gene-expression regulation during the exit from naïve pluripotency. Genes exhibiting parentally biased allele-specific expression, equal expression from both alleles, published imprinted genes and all genes, were used as gene groups to compare the dynamics in gene expression as stated in the text.

**Whole-genome bisulfite sequencing.** Sequencing libraries for DNA methylation mapping were prepared using the µWGBS protocol[24]. Starting directly from lysed cells in digestion buffer, proteinase K digestion was performed at 50 °C for 20 minutes. Custom-designed methylated and unmethylated oligonucleotides were added at a concentration of 0.1% to serve as spike-in controls for monitoring bisulfite conversion efficiency. Bisulfite conversion was performed using the EZ DNA Methylation-Direct Kit (Zymo Research, D5020) according to the manufacturer's protocol, with the modification of eluting the DNA in only 9 µl of elution buffer. Bisulfite-converted DNA was used for single-stranded library preparation using the EpiGnome Methyl-Seq kit (Epicentre, EGMK81312) with the described modifications. Quality control of the final library was performed by measuring DNA concentrations using the Qubit dsDNA HS assay (Life Technologies, Q32851) on Qubit 2.0 Fluorometer (Life Technologies, Q32866) and by determining library fragment sizes with the Agilent High Sensitivity DNA Analysis kit (Agilent, 5067-4626) on Agilent 2100 Bioanalyzer Station (Agilent, G2939AA). All libraries were sequenced by the Biomedical Sequencing Facility at CeMM using the 2x75bp paired-end setup on the Illumina HiSeq 3000/4000 platform.

**DNA methylation data processing.** Sequencing adapter fragments were trimmed using Trimmomatic v0.32[65]. The trimmed reads were aligned with Bismark v0.12.2[66] with the following parameters: --minins 0 --maxins 6000 --bowtie2, which uses Bowtie2 v2.2.4[67] for read alignment to the mm10 assembly of the mouse reference genome. Duplicate reads were removed as potential PCR artefacts and reads with a bisulfite conversion rate below 90% or with fewer than three cytosines outside a CpG context (required to confidently assess bisulfite conversion rate) were removed as potential post-bisulfite contamination. DNA methylation levels estimated by the Bismark extractor were loaded into R retaining all CpGs that were covered with at least three reads in at least two samples. We then used dmrseq[30] (v1.6.0) to identify consistently methylated regions of neighbouring CpGs ($n = 168,061$ regions) between androgenote, parthenogenote, and ICSI blastocysts (two replicates per sample group, total $n = 6$). We retained all regions with opposing DNA methylation levels in uniparental vs ICSI blastocysts (ie either aha > ICSI > pha, or aha < ICSI < pha), with at least 100 reads total coverage (across all replicates), and with a minimum length of 100 bp. Testing those regions ($n = 77,358$) for significant differences in DNA methylation levels by sample group (FDR-adjusted p-value ≤ 0.1, |aha-pha| ≥30 percentage points, |β_{aha}| ≥0.25, |β_{pha}| ≥ 0.25) yielded 859 candidate DMRs. To enable comparison of these DMRs with the DNA methylation status in oocytes, sperm, and the ICM, we obtained published MethylC-Seq data (Wang et al., 2014) from GEO (GSE56697) and performed hierarchical clustering with complete linkage.

**Positional, region overlap and motif enrichment analysis.** We examined DMR regions using two complementary approaches. First, Locus Overlap Analysis[32] (LOLA; v1.16.0) identified significant overlaps with experimentally determined transcription factor binding sites from publicly available ChIP-seq data. We used 791 ChIP-seq peak datasets from the LOLA Core database (version 180423) and added Znf445 binding peaks[36]. We considered all terms with an 8-fold enrichment and an FDR-adjusted p-value below 0.005 significant. We focused on datasets from ESCs. Secondly, we searched DNA sequences of each DMR for matches to known DNA-binding motifs from the HOCOMOCO database v11[37] (using only motifs with quality score A or B). For this, we used FIMO[68] (v4.10.2) (parameters: --no-qvalue --text --bgfile motif-file), and regions with at least one hit ($p < 0.0001$) were counted. To test for motif enrichment, we used Fisher's exact test. Motifs with an 8-fold enrichment and an FDR-adjusted p-value below 0.005 were considered significant.

**Analysis of allele-specific H3K27me3.** We defined parent-of-origin-specific H3K27me3 imprints using published allele-specific ChIP-seq data from the ICM and gametes[43]. To this end, we downloaded peak coordinates from GEO (GSE76687) and converted them to the mm10 genome assembly using liftOver. We considered a gene to be linked to allele-specific H3K27me3 if a peak from the ICM dataset was found within 5 kb of its transcription start site. For gametes, a gene was considered oocyte- or sperm-specifically marked by H3K27me3 if an associated peak (GSE76687_MII_oocyte_k27me3_broadpeak.bed, GSM2041066_Sperm_H3K27me3_broadpeak.bed) was found in only one dataset. For the ICM, we used published parent-of-origin-specific peak lists from GEO and considered a gene to be parent-of-origin-independently H3K27me3-associated ('both') if an associated H3K27me3 peak (GSE76687_ICM_k27me3.bed.gz) was found but not a maternal- or paternal-specific H3K27me3 peak (GSE76687_ICM_K27me3_maternal.bed, GSE76687_ICM_K27me3_paternal.bed).

**Analysis of DNA-methylation-dependent and H3K27me3-dependent allelic expression.** To assess dependence of the allelic expression bias of imprinted genes on DNA methylation and H3K27me3, we obtained allele-specific RNA-seq data before and after knockout (KO) of maternal Dnmt3l (mDnmt3l) and Eed (mEed)[45,46] from GEO (GSE130115 and GSE116713). To exclude bias, we first confirmed that no differentially expressed genes were found between WT and KOs using DESeq2 (independent of the allelic expression status; FDR-adj. p ≤ 0.1; v1.26.0). We then tested all BsX and published imprinted genes for parent-of-origin-specific expression by comparing maternal and paternal read counts in WT samples using DESeq2. For significantly parent-of-origin-specifically expressed genes (FDR-adj. p ≤ 0.1), we performed a second test to evaluate the dependence of the allelic ratio on the genotype (WT vs KO). Genes with a significant parent-of-origin-specific expression (FDR-adj. p ≤ 0.1) significantly dependent on the genotype (FDR-adj. p ≤ 0.05), that showed a reduction in their allelic bias in KO compared to WT sample, were defined as 'dependent' on either mDnmt3l or mEed. To call mEed dependency, we only considered genes expressed with paternal bias in the WT sample.

**E3.5 and E6.5 embryo RNA extraction, RT-PCR and Sanger sequencing.** We used the cDNA from SMART-Seq2 libraries independently generated from libraries used for RNA-seq analysis as a template to amplify PCR fragments covering at least one SNP per gene. Resulting fragments were then analysed by Sanger sequencing. Primers used for PCR amplification and Sanger sequencing analysis are reported in Supplementary Table 1.

**Topologically associating domains (TAD).** We employed TADs coordinates from the 64-cell stage[42] as a proxy for E3.5 blastocysts. TADs finding and filtering (TAD-score ≥ 0.5) was performed as in[42]. We used identified TAD coordinates to determine if genes are associated with at least one of our 859 DMRs within the same TADs. DMR and imprinted genes are defined to be within the same TAD if the centre of DMR and gene transcription start sites are located within the same TADs. The control set was generated by randomly shifting genome coordinates of both genes and DMR on each chromosome 2,000 times, in a manner that maintained identical pair-wise genomic distances between genes and DMRs (i.e., the DMR-gene distances in control and sample set are the same). This random control shows how frequently DMR-BsX co-occurrence is expected by chance. The number of imprinted genes (BsX) having at least one DMR associated within the same TAD was calculated and compared to the random control to obtain empirical p-values.

**Reporting summary.** Further information on research design is available in the Nature Research Reporting Summary linked to this article.

## Data availability

RNA-seq and µWGBS data generated in this study were submitted to the Gene Expression Omnibus (GEO, GSE152106). All other relevant data supporting the key findings of this study are available within the article and its Supplementary Information files or from the corresponding authors upon reasonable request. Source data are provided with this paper.

A reporting summary for this article is available as a Supplementary Information file. Source data are provided with this paper.

## Code availability

Custom computer code used for data analysis is available from https://github.com/cancerbits/santini2021_imprints.

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

## Acknowledgements

The authors thank Robert Feil and Anton Wutz for helpful discussions and comments, Samuel Collombet and Peter Fraser for sharing embryo TAD coordinates, and Andy Riddel at the Cambridge Stem Cell Institute and Thomas Sauer at the Max Perutz Laboratories FACS facility for flow-sorting. We thank the team of the Biomedical Sequencing Facility at the CeMM and the Vienna Biocenter Core Facilities (VBCF) for support with next-generation sequencing. We are grateful to animal care teams at the University of Bath and MRC Harwell. A.C.F.P. acknowledges support from the UK Medical Research Council (MR/N000080/1 and MR/N020294/1) and Biotechnology and Biological Sciences Research Council (BB/P009506/1). L.S. is part of the FWF doctoral programme SMICH and supported by an Austrian Academy of Sciences DOC Fellowship. M.L. is funded by a Vienna Research Group for Young Investigators grant (VRG14-006) by the Vienna Science and Technology Fund (WWTF) and by the Austrian Science Fund FWF (I3786 and P31334).

## Author contributions

L.S. performed experiments and analysed data and F.H. performed biocomputational data analysis. F.T.-T. performed allele-specific RNA-seq analysis with support from F.P. and S.H., supervised by A.B. T.S. and A.C.F.P. generated androgenetic and partheno-genetic embryos and performed semicloning experiments. M.A. performed embryo immunofluorescence assays. N.W. prepared E6.5 embryos. M.F. and C.B. performed and supervised SMART-seq2 and µWGBS analysis. J.R. provided experimental support to L.S. A.L. performed PCR analysis and X.M. performed TAD analysis supervised by E.D.L. A.C.F.P. and M.L. conceived and supervised the study, provided funding and analysed data. A.C.F.P. and M.L. wrote the paper together with L.S., F.H. and F.T.-T. and input from all authors.

## Competing interests

The authors declare no competing interests.
