## [Peer Review File · Nature Communications]

Reviewer Comments

Reviewer #1 (Remarks to the Author):

See attached PDF

Reviewer #2 (Remarks to the Author):

In this manuscript, the authors (Santini et al.) sought to understand which mechanisms or pathways control gene imprinting in mouse blastocyst stage. Toward this end, they first identified 71 novel imprinted genes, by RNA-sequencing the hybrid mouse embryos, calling the allelically skewed gene expression, and intersecting with public resources of mouse imprinting genes. Next, they performed the micro-whole-genome bisulfite sequencing (μ WGBS) of individual uniparental blastocysts to examine if the imprinted genes are associated with allelic DNA methylation. Further, they took the published H3K27me3 data to examine whether the DNA methylation-independent imprinted genes are associated with H3K27me3 marks. Finally, they used the maternal Dnmt3l KO or Eed KO datasets to validate their results. The analysis is decent, and the manuscript is written well. However, I have several major and minor concerns.

Major points:

- (1) The discovery and methods present in this study are very similar to that in Inoue et al, 2017. What are the main advantages and discoveries in this study? That will be great if the authors can clarify it and make it clearer to the reader.
- (2) Given that the imprinting status of genes depends on the cell types and developmental stages, I just wonder whether the Dnmt3l KO and Eed KO dataset used in the analysis matches the developmental stages in which the authors characterised gene imprinting.
- (3) In Line 79, the authors reported that 100 published imprinted genes were absent from the BiX dataset, including Igf2, H13, and Commd1. However, in Line 85, they tried with different statistical methods and only found 24 out of 134 published imprinted genes show significant biallelic expression. What about the rest 76 genes (100 - 24)?
- (4) Line 235, the authors said "the vast majority of both nBiX and nBsX genes exhibited paternal expression with maternal 5mC at the closest DMR, similar to published imprinted genes". However, in Line 239, they added that "nBiX and nBsX did not show such as association". These two claims seem opposite.

Minor questions:

- (1) Line 58, the authors excluded the potential X-linked imprinted genes. Can the authors state the reason why X-linked genes are removed from the analysis? Mixed gender embryos? X chromosome inactivation? Or other reasons.
- (2) Line 76, missing the second bracket.
- (3) In Fig 1, the authors reported that expression of Commd1 in blastocytes showed mixed pattern, i.e. some are biallelic expression while some are maternal- or paternal specific expression pattern. Can the authors add some details about the original study that Commd1 is imprinted and which tissues or developmental stage they used, to make the readers clear about the discrepancy.
- (4) In extended Fig 1g, the authors should specify what "KO" stands for. Do these "KO" data here support this paper's conclusion?

- (5) Line 210-215, there are 859 DMRs in total, including 410 from clusters C1 and C4, 62 from cluster C5, 349 from cluster C2, and 37 from cluster C5. $410+62+349+37=858<859$.
- (6) Line 214, the authors said that “most differential DNA methylation is encoded within gamete genomes”. This claim is not appropriate, because cluster C2 and C3 sites are re-established group during preimplantation development that nearly account for 45%.
- (7) I appreciate the authors tried to provide much information in the figure 3d, but it's very difficult to understand and the legend is not clear.
- (8) Line 217, “overlapped” should be “overlap”.
- (9) Line 335, the authors found that in some cases the differential DNA methylation and parent-of-origin-specific expression is unlinked. I am wondering whether it depends on the location of this DNA methylation in gene body or promoter. For example, gene body DNA methylation doesn't link to the allelic expression and promoter methylation does.
- (10) That will be great if the authors can add the parent-of-origin (maternal or paternal) for imprinting genes to their figure 6.

Reviewer #3 (Remarks to the Author):

In their manuscript ‘Novel imprints in mouse blastocysts are predominantly DNA methylation independent’, Santini et al. present a multi-layered analysis of the imprinting landscape in mouse preimplantation embryos. Their study follows previous work demonstrating that some imprints are transient during this stage and do not depend on parent-specific DNA methylation as classical imprints do, but rather rely on maternally-inherited H3K27me3—a phenomenon that has been termed ‘non-canonical imprinting’ (Inoue et al., Nature 2017). Here, the authors sought to determine the extent to which each of these mechanisms contributes to the regulation of imprinting in the early embryo. By employing allele-specific RNA-seq of reciprocal F1 hybrids, DMR profiling of uniparental and biparental embryos, and analysis of published ChIP-seq and Hi-C datasets, they identified more than 70 putatively imprinted genes, most of which are associated with parent-of-origin-specific H3K27me3 rather than a DMR. This is a carefully executed and comprehensive study that highlights the impact that different regulatory layers may have on imprinting. However, the following issues should be addressed before this manuscript can be considered for publication.

Major comments

1. Studying the dynamics of imprinting regulation, which is uniquely stable in early embryogenesis despite immense changes to the epigenome, is important due to its dire implications on development. Although the findings of this work argue in favor of a predominant role for H3K27me3 in regulating imprinting during early development, it is still unknown whether the observed parent-specific expression patterns are functionally important or if they merely reflect a byproduct of the numerous differences between the parental genomes at these stages. This is a fundamental point that should be discussed.
2. Previously described ‘non-canonical imprinted genes’ are thought to be exclusively paternally-expressed due to maternally-derived H3K27me3. However, some of the genes reported as novel BiX genes (such as *Emc2*) were found to be maternally-expressed, suggesting paternally-derived H3K27me3. This point should be studied further.
3. A crucial aspect of identifying novel imprinted genes is demonstrating not only parental-allele-biased expression but also parent-of-origin epigenetic inheritance. Therefore, the authors should analyze H3K27me3 in oocytes and sperm to determine whether the parent-

specific patterns observed in embryos are inherited from the gametes (complementing the analysis of DMRs in Fig 3c).

4. The authors should elaborate on how many of the 40% of novel BiX/BsX genes that depend on either maternal DMRs or H3K27me3 based on KO morulae (Fig. 5e), depend on each or both mechanisms. These data are directly linked to the overall aim and conclusion of the paper.

5. How many of the detected BiX/BsX genes were already identified by previous studies reporting non-canonical imprinting? It is important to discuss these numbers as an indication for reproducibility, as well as for the feasibility to identify new genes regulated by this putative mechanism.

6. The monoallelic expression of several of the novel BiX/BsX genes were confirmed by Sanger sequencing. How many of these novel genes were tested in total, and what was the false discovery rate?

7. The data in Fig. 5a–c should be accompanied by a statistical analysis to help determine the extent of similarity between the observed molecular phenotypes in control vs. KO embryos. It would also be helpful to explain this complex Figure in more detail in the text and/or in the Figure legend.

8. The conclusions drawn from Fig. 1h (starting from page 7, last paragraph) are not supported by statistically significant results.

Minor comments

1. The claim made in the Abstract that “neither parent-of-origin-specific transcription nor DMRs have been comprehensively mapped” is untrue. There is a multitude of papers that have done so and should be cited (to name a few: Court et al., *Genome Res.* 2014; Babak et al., *Nat. Genet.* 2015).

2. The term ‘blastocyst-imprinted expressed’ (BiX) genes was defined based on parent-of-origin allele-specific expression alone. However, to qualify as truly imprinted, parent-specific epigenetic marks need to be identified. Therefore, this set of genes should be regarded as ‘potentially’ or ‘putatively’ imprinted.

3. The statement that somatic DMRs “carry marks at the blastocyst stage that are not directly dependent on DNA-methylation” (page 9, starting from line 155) is not entirely accurate. For example, the Meg3 somatic DMR depends on differential methylation of the IG-DMR at this locus.

4. In “thirty-seven blastocyst DMRs exhibited little or no DNA methylation in oocytes or sperm” (line 212), the percentage out of all DMRs should be indicated to highlight this cluster of genes represents a minority of DMRs.

5. The intriguing finding that “confirmed published imprinted genes showed only limited response to loss of mDnmt3l” (line 285) should be discussed.

6. In “expression of some 50% of novel imprinted genes were apparently H3K27me3- and

DMR- independent” (line 303), it should be clarified that this relates only to maternal H3K27me3 and DMRs.

7. Figure improvements:

- What are the WT/KO genotypes in Extended Fig. 1g?
- I found Fig. 3d–f challenging to interpret. Also, it is unclear what the ‘+’ in parentheses represent
- Fig. 5a–c are very important but hard to grasp. Consider simplifying them
- Fig. 7 could be improved by adding close-up views of representative clusters

This manuscript, authored by Laura Santini and Florian Halbritter *et al*, is entitled “Novel imprints in mouse blastocysts are predominantly DNA methylation independent”. Using combined approaches of transcriptomics and DNA methylation profiling in mouse blastocysts, haploid androgenote and parthenogenote blastocytes and mESCs, the authors have evidenced new parentally skewed (nBsX) or imprinted (nBiX) genes and new DMRs. Furthermore, most of the newly found nBsX and nBiX genes seemed independent of DNA methylation but linked to H3K27me3 maternal enrichment.

This study is interesting and use elegant approaches of transcriptomics, chromatin profiling and 3D conformation to study the regulation of newly found imprinted or skewed loci at the blastocyst stage. Fine regulation of gene expression at this stage is critical for proper embryo development and misregulation of imprinted genes specific of this stage can have long life effect such as with the *Liz/Zdbf2* locus.

This study can be of importance but is lacking some crucial analysis, such as the link between these nBsX and nBiX genes and the polycomb inherited domains (Collombet *et al*, 2020 Nature and Du *et al*, 2020, Molecular Cell from respectively the Heard lab and Xie lab), see below major points. Furthermore, an effort needs to be made for better presentation of the results as well as improved description of their experiments and analysis in the text and figure legends.

Major points:

- Are these imprinted genes lost upon development? Are they lowly expressed genes? Are they important for early differentiation, extraembryonic tissue or late somatic differentiation? Please develop the analysis of these nBsX and nBiX.
- There is a bias for B6 expressed genes compared to Cast. Could that be explained by C57Bl/6J being the reference genome and how the reads are aligned to genomes?
- How did the authors chose the threshold of 70/30 expression ratio to call a gene imprinted? It does not seem very stringent to talk about imprinted genes.
- The study of DNA methylation erasure in the diploid and haploid cell line is interesting, however several criteria could impact the degree of DMR erasure (Figure 2) such as number of passages, sex or hybrid genetic background of the ESCs. The authors should check if some of these criteria could at least partially explain the different phenotype of hypomethylation and discuss it in the manuscript.
- The enrichment search for DNA binding motifs gives interesting insights. Is there any difference between the groups, *eg* parentally inherited DMRs versus the *de novo* ones?
- For the 3D topological domains and the TADs study:
 - 1) The authors should not assume conservation of TADs (line 254) upon preimplantation development when latest publications have highlighted different domains after fertilization that are quickly lost and only after apparition of TADs (around morula stage).
 - 2) It is important that the authors do their analysis with the hybrid HiC data from the Collombet *et al*, 2020 study and the Du *et al*, 2020 during preimplantation development. Both papers have highlighted novel Polycomb domains that are inherited from the parental gametes and conserved in the first cleavage stages. These domains correlate with gene silencing in early embryos and novel H3K27me3 dependent imprinted genes. The authors should then compare their nBiX and nBsX genes to these Polycomb parentally-inherited genes to better understand the mechanisms of these finely regulated genes.

Minor Points

-Line 55

C57Bl/6J is a *Mus musculus domesticus* derived strain and not *M. m. musculus*.

-Figure 1A

Please avoid abbreviations if not explained in the figure legends. Which cross is the reverse cross (RV?) with n=5 single blastocysts? B6 x Cast?

This figure is not very informative as it is.

-Figure 1C

Scale bar is missing information.

-Extended Figure 1 G:

Use of published single cell datasets, WT and KO. What is the KO for? It should be explained clearly at least in the figure legend which gene is knock-out.

-Figure 1h

Data processing needs better explanation. Why Eed and Nanog/Oct4 have been highlighted from the results for the nBiX genes? Is that significant?

- Figure 2b: cell line called ahES but written aES. It is not easy to follow ahES and pHES abbreviations as usually hES stands for human ESCs.

- Line 125 Please be careful of overstatement and change the following sentence accordingly.

"This indicates that genes showing parent-of-origin-specific gene expression in blastocysts are involved in, or responsive to early embryonic cell fate specification."

- Extended figure 2c:

I did not understand how the datasets have been produced. It should be better introduced to be comparable.

-Line 203

Only the Liz1/Zdbf2 GL-DMR was not confidently identified in our analysis, because it lacked a DNA methylation signal in one of the ICSI samples (**Fig. 3a**).

I don't understand how this statement can be extracted from Figure 3a?

-Figure3:

Why E4.5 candidates? According to the manuscript, blastocysts were collected at E3.5. The authors should better explain how the embryos have been obtained, processed and analyzed.

-Extended figure 3a is unreadable.

-Figure 4:

What is the high confidence repository imprints? What is the difference with the 30 confirmed in blastocyst?

-Figure 6:

According to the text, all the novel clusters should integrate nBiX or nBsX genes (in bold) but it is not the case of clusters 1 to 23. So what are these clusters and how have they been selected?

Reviewer #1

This manuscript, authored by Laura Santini and Florian Halbritter et al, is entitled “Novel imprints in mouse blastocysts are predominantly DNA methylation independent”. Using combined approaches of transcriptomics and DNA methylation profiling in mouse blastocysts, haploid androgenote and parthenogenote blastocytes and mESCs, the authors have evidenced new parentally skewed (nBsX) or imprinted (nBiX) genes and new DMRs. Furthermore, most of the newly found nBsX and nBiX genes seemed independent of DNA methylation but linked to H3K27me3 maternal enrichment.

This study is interesting and use elegant approaches of transcriptomics, chromatin profiling and 3D conformation to study the regulation of newly found imprinted or skewed loci at the blastocyst stage. Fine regulation of gene expression at this stage is critical for proper embryo development and misregulation of imprinted genes specific of this stage can have long life effect such as with the *Liz/Zdbf2* locus.

This study can be of importance but is lacking some crucial analysis, such as the link between these nBsX and nBiX genes and the polycomb inherited domains (Collombet et al, 2020 Nature and Du et al, 2020, Molecular Cell from respectively the Heard lab and Xie lab), see below major points. Furthermore, an effort needs to be made for better presentation of the results as well as improved description of their experiments and analysis in the text and Figure legends.

We thank the reviewer for the overall positive assessment of our study. We have added data and extensive analyses, including those based on TADs present during embryo development. We have worked to improve presentation throughout the manuscript, as detailed below.

Major points:

- Are these imprinted genes lost upon development? Are they lowly expressed genes? Are they important for early differentiation, extraembryonic tissue or late somatic differentiation? Please develop the analysis of these nBsX and nBiX.

Parent-of-origin expression of nBsX genes is lost between E3.5 and E6.5. This is described in the text (l.99 and in Fig. 1d,f and Supplementary Fig. 1h,i).

We have taken measures to ensure fidelity of our datasets by excluding genes too lowly expressed or containing too few SNPs for reliable analysis. New Figure R1 shows that overall, the expression levels of identified nBsX and confirmed published imprinted genes (pubBsX; these are genes that were previously reported to be imprinted which show parent of origin specific expression in our datasets; see main text for details) are not significantly different from each other. The Figure also shows that nBsX genes do not belong to a group of lowly expressed genes.

Figure R1. Boxplot showing Base Mean values (from the RNA-Seq analysis performed in our study) for different gene groups (All detected genes, nBsX, pubBsX, and published unconfirmed genes). *, $p < 0.05$; **, $p < 0.01$; ***, $p < 0.001$; ns, not significant.

In raising these issues, the Reviewer flags the important question of whether the nBsX/nBsX genes belong to a specific functional group. To address this, we have now performed two additional analyses:

First, to evaluate roles of blastocyst imprinted genes in early differentiation decisions, we investigated how many nBsX genes and published imprints are found associated with exit from naïve pluripotency in three independent genome-wide screens (Leeb et al., 2014; Vilegas et al., 2018 [Betschinger lab]; Li et al., 2018 [Yusa lab]). As shown in Figure R2, this analysis revealed that only a minor fraction of candidate genes involved in exit from naïve pluripotency also exhibited parentally biased expression, and *vice versa*. Therefore, we conclude that neither published, nor novel imprinted genes, as a group, play a major role in the exit from naïve pluripotency.

Secondly, we asked whether we can detect tissue-specific expression of sets of published and novel imprinted genes. To this end, we used the Bioconductor package, TissueEnrich, to detect tissue-specific expression patterns of nBsX genes, based on ENCODE transcription data.

Figure R2. Venn diagrams showing overlaps between published candidates in exit from naïve pluripotency screens, published imprinted genes and BsX genes.

This analysis revealed that nBiX and nBsX genes exhibit tissue- and stage-specific expression in the E14.5 mouse brain. This finding is now presented in Figure 1g (reproduced in Fig. R2, below) and described in the main text (l.116-123).

Delineating the roles of novel imprinted genes is beyond the scope of this study, but specific expression in the mid-gestation brain forms the basis of testable hypotheses in future work.

Figure R3. (See also manuscript in Fig 1g.) Barplot showing Tissue-specific gene enrichment for different gene groups (nBiX, nBsX, pBsX, Published unconfirmed and equivalent genes). Only Tissues with a significant (adj. $p < 0.05$) enrichment in at least one group of genes are shown. *, adj. $p < 0.05$; **, adj. $p < 0.01$; ***, adj. $p < 0.001$; ns, not significant.

- There is a bias for B6 expressed genes compared to Cast. Could that be explained by C57Bl/6J being the reference genome and how the reads are aligned to genomes?

This is a good point and we agree that strain-specific bias can potentially be generated by differential mapping efficiencies, as observed by the authors of the allelome.pro pipeline employed in this work (Andergassen et al., *Nucl. Acids Res.* [2015]; <https://doi.org/10.1093/nar/gkv727>). Specifically, the STAR aligner was found to result in maximum discovery of SNPs, with similar allelic skewing to other mappers such as, for example, *GNAS*, which was evaluated for comparison in the cited paper. This prompted us to use STAR aligner. However, we agree with the inference that mapping to B6 as reference sequence may indeed contribute to the skewing observed toward the B6 alleles. Accordingly, we have now removed our analysis pertaining to strain-specific expression.

Importantly, by using reciprocal crosses and stringent statistics and cutoffs that require consistent allelic skewing in both cross-directions, the discovery of parent-of-origin-specifically expressed genes is not affected by potential strain-biased mapping.

- How did the authors choose the threshold of 70/30 expression ratio to call a gene imprinted? It does not seem very stringent to talk about imprinted genes.

The 70:30 cutoff has been adopted by others, and we now cite a reference that illustrates this approach (Andergassen et al., 2017) and which shows that many known and well-studied imprinted genes fall within this range of allelic expression. We were also advised on this point in discussions with imprinting experts including Prof. Robert Feil (IGMM, Montpellier, France). Our data show that using this cutoff, we were able to capture published imprinted gene expression and identify 71 additional genes that exhibited previously unreported parent-of-origin-specific expression. The high validation rate by RT-qPCR followed by Sanger

sequencing in independently-derived blastocysts (I.105) corroborates the suitability of this cutoff.

- The study of DNA methylation erasure in the diploid and haploid cell line is interesting, however several criteria could impact the degree of DMR erasure (Figure 2) such as number of passages, sex or hybrid genetic background of the ESCs. The authors should check if some of these criteria could at least partially explain the different phenotype of hypomethylation and discuss it in the manuscript.

Thank you for raising this point. We agree that cell line passage number, culture conditions and sex have the potential to influence DNA methylation levels. We now highlight this in the manuscript (I.160). Our datasets include male and female ESCs at different passage numbers, although we were unable to detect a clear pattern regarding DNA methylation levels (Fig. R4). The data suggest that although diploid ESCs largely maintain a 'normal' imprinted DNA methylation pattern, haploid ESCs are prone to loss of DMRs. That said, the experiment was not designed to support strong claims about differences in sex as it is a relatively minor aspect of the work and we have taken care not to over-state its significance in the manuscript. As far as these analyses go, they suggest that there is no absolute correlation between sex, passage number and strength of methylation over DMRs.

Figure R4. Plot showing overall methylation levels across DMRs in indicated cell lines and correlation between parental provenance, sex and passage number to methylation levels over DMRs identified in this study. The data indicate a reduction of DMR methylation in parthenogenetic ESCs at higher passage numbers and an overall loss of DMR methylation in androgenetic haploid ESCs. Lines indicate the mean per indicated sample group and passage number, as a visual guide.

- The enrichment search for DNA binding motifs gives interesting insights. Is there any difference between the groups, eg parentally inherited DMRs versus the de novo ones?

Thank you for this comment, in response to which we have extended Figure 2d-f to include DMR clusters identified in Figure 2c and increased their depth of analysis; an excerpt is shown below (Fig. R5). Please note that we renamed the clusters from top to bottom (DMR-C1 to DMR-C5) in an effort to make the manuscript easier to follow; this means that cluster numbers in the first submission do not match those in the figure below. In general, we find little difference between DMR clusters identified in Figure 2c in terms of TF motif or ChIP binding enrichment. The exceptions are overlaps with binding sites (ChIP-seq peaks) of Polycomb group members Ezh2, Suz12, and Rnf2 (and to a lesser degree Jarid2), which are enriched in DMR clusters C1 and C2 (which both represent maternal imprinting maintained from oocytes), although they were less enriched, or not enriched at all, in the other clusters (Fig. 2e).

e

f

Figure R5. Excerpt from Figure 2; panels e and f. The background of all detected methylated sequences is shown in grey. DMR-X refers to DMR clusters identified in Figure 2c. e, Locus overlap analysis (Sheffield et al., 2015) of published ChIP peaks on known GL-DMRs and novel DMRs. f, Motif enrichment analysis (Grant et al, 2011; Kulakovskiy et al., 2017) in known GL-DMRs and novel DMRs

- For the 3D topological domains and the TADs study:

1) The authors should not assume conservation of TADs (line 254) upon preimplantation development when latest publications have highlighted different domains after fertilization that are quickly lost and only after apparition of TADs (around morula stage).

2) It is important that the authors do their analysis with the hybrid HiC data from the Collombet et al, 2020 study and the Du et al, 2020 during preimplantation development. Both papers have highlighted novel Polycomb domains that are inherited from the parental gametes and conserved in the first cleavage stages.

These domains correlate with gene silencing in early embryos and novel H3K27me3 dependent imprinted genes.

The authors should then compare their nBiX and nBsX genes to these Polycomb parentally-inherited genes to better understand the mechanisms of these finely regulated genes.

We are grateful to the Reviewer for raising this point and completely agree that the dynamic nature of TAD organization in preimplantation embryos should be considered, which we have now attempted to do.

Du et al, 2020 use ICM data that they had published previously (Du *et al.*, 2017) and state that TAD boundaries in the ICM largely overlap with TAD boundaries in ESCs (doi.org/10.1038/nature23263; see Extended Data Fig. 4). Taking on board the point made by Reviewer #1, we therefore decided to utilize the more recent datasets from Collombet *et al.* (2020), who have identified dynamic reorganization of TAD-structure during preimplantation development. To achieve this, we contacted the authors and have now repeated our analysis with data provided by them from their paper (Collombet *et al.*, 2020). These data contributed to Figure 4b and Supplementary Figure 4d, Figure 5f, g and Supplementary Figure 5e, Figure 6a,b and Supplementary Figure 6a,b. In terms of the number of BsX and BiX genes co-localising with a DMR in a TAD, these new analyses using 64C embryo data and previous data using ESC TADs give similar results over all, and our previous conclusions are unaffected.

Collombet *et al.* further identified three clusters of TADs that are specifically regulated by Polycomb activity during preimplantation development. We have determined that 30% and 32% nBiX and nBsX genes, respectively, and 47% of pubBsX genes are part of these clusters (Fig. R6), supporting the notion that the Polycomb system plays a major role in regulating imprinted gene expression.

Figure R6 Heatmap showing the presence of genes from indicated groups in Polycomb-associated TAD clusters defined by Collombet et al. (2020).

We have now also further addressed the question of how H3K27me3-dependent parent-of-origin-specific gene expression is inherited. To this end, we added analyses utilizing allele specific H3K27me3 datasets from sperm, oocytes and ICM. For ~50% of genes within the paternally expressed group of nBsX, we identified H3K27me3 at the TSS in oocytes, which was maintained and even slightly increased in the ICM (see manuscript Fig. 4 and Fig. R7, below). Even within the group of published confirmed imprinted genes, we show that the TSSs corresponding to most (~60%) were decorated with H3K27me3.

Figure R7. (See Fig. 4, manuscript) **a** Heatmap showing association between BsX genes and allele-specific (in the ICM at the blastocyst stage) or gamete-specific H3K27me3. Color codes distinguish between allelic expression of the BsX genes (maternal or paternal), allelic presence of the H3K27me3 mark (on paternal or maternal allele in the ICM, in sperm or in oocyte), and different gene groups (published confirmed genes, nBiXs or nBsXs). **b**, Pie charts illustrating the occurrence of ICM allele-specific or gamete-specific H3K27me3 at the TSS of all 10,743 robustly detected genes. **c** and **d**, Pie charts illustrating the occurrence of ICM allele-specific or gamete-specific H3K27me3 at the TSS of (c) maternally or (d) paternally expressed nBiX, nBsX and published confirmed genes.

Minor Points

-Line 55, C57Bl/6J is a *Mus musculus domesticus* derived strain and not *M. m. musculus*.

We apologise for this mistake and thank the reviewer for pointing it out; it is now corrected.

-Figure 1A, Please avoid abbreviations if not explained in the Figure legends. Which cross is the reverse cross (RV?) with n=5 single blastocysts? B6 x Cast? This Figure is not very informative as it is.

We now provide information explaining all abbreviations, including FW and RV in the legend to Supplementary Figure 1a.

-Figure 1C, Scale bar is missing information.

We have added a scale bar (Fig. 1b). The colour code shows the relative maternal:paternal expression. The y-axis shows expression relative to the maximum.

-Extended Figure 1 G: Use of published single cell datasets, WT and KO. What is the KO for? Tt should be explained clearly at least in the Figure legend which gene is knock-out.

KO refers to an *Xist* knock out, as now explained in the Figure legend and Methods. Expression profiles in WT and KO single cells for nBiX and other tested gene groups is very similar (X-linked genes were excluded from our analysis to ensure comparability with our own datasets).

-Figure 1h, Data processing needs better explanation. Why Eed and Nanog/Oct4 have been highlighted from the results for the nBiX genes? Is that significant?

We agree and apologize for previously including these data, which are uninformative. They have now been replaced. For a more informative portrayal of the role of (n)BiX genes during development, we instead determined whether BiX and BsX genes are specifically enriched in certain tissues based on ENCODE expression data: nBiX genes are enriched for genes expressed in E14 brain. These data are now presented in Figure 1g (see also the first point, above).

- Figure 2b: cell line called ahES but written aES. It is not easy to follow ahES and phES abbreviations as usually hES stands for human ESCs.

We now use ahaESC for androgenetic, and phaESC for parthenogenetic haploid mouse ESCs wherever we refer to them.

- Line 125 Please be careful of overstatement and change the following sentence accordingly. "This indicates that genes showing parent-of-origin-specific gene expression in blastocysts are involved in, or responsive to early embryonic cell fate specification."

We have deleted this statement.

- Extended Figure 2c: I did not understand how the datasets have been produced. It should be better introduced to be comparable.

This plot is now presented as Supplementary Figure 2e (see Fig. R8). We have modified the plot so that DMRs not detected in allele-specific ICM analysis are presented. This shows that for differential methylation detected in gametes, SNP-based deconvolution of DMR-associated reads only identified 6 of the 24 known DMRs.

Figure R8. (See also Supplementary Figure 2e.) Heatmap showing DNA methylation signals for 24 known GL-DMRs in ICM samples from previous data (Wang et al., 2014), distinguishing between maternal and paternal alleles. Grey boxes with an 'X' indicate no data. The colour scale represents percentage of 5mC compared to 5C.

-Line 203, Only the Liz1/Zdbf2 GL-DMR was not confidently identified in our analysis, because it lacked a DNA methylation signal in one of the ICSI samples (Fig. 3a). I don't understand how this statement can be extracted from Figure 3a?

Thank you for pointing out this error. The figure reference was indeed incorrect and should have been to the previous figure, Figure 2c, which is now Figure 2a. This point has now been corrected.

-Figure3: Why E4.5 candidates? According to the manuscript, blastocysts were collected at E3.5. The authors should better explain how the embryos have been obtained, processed and analyzed.

We apologize for causing confusion with this error and the embryo stage has now been corrected to E3.5 throughout the manuscript.

-Extended Figure 3a is unreadable.

Thank you for pointing this out. We have revised this plot in an effort to optimise its clarity, and now show it as Supplementary Figure 2j.

-Figure 4: What is the high confidence repository imprints? What is the difference with the 30 confirmed in blastocyst?

We thank the Reviewer; we should previously have been clearer. We classify the 30 genes that have been reported in three or more publicly available imprinting repositories as high confidence (HCon) repository imprinted genes, regardless of their expression state in our analysis. In total we confirmed parent-of-origin biased expression for 36 out of 134 published imprinted genes and 10 out of 30 HCon repository genes. This is now explained in the text (l.63-65).

-Figure 6: According to the text, all the novel clusters should integrate nBiX or nBsX genes (in bold) but it is not the case of clusters 1 to 23. So what are these clusters and how have they been selected?

We have redesigned Figure 6 and Supplementary Figure 6 and now show genome snapshots for all identified clusters. Not all imprinting clusters contained nBiX or nBsX genes. For clustering, we included all nBsX genes as well as all published imprinted genes, regardless of their expression status at the blastocyst stage. Resulting clusters were then categorized according to whether they contained only published imprinted genes, clusters where nBsX genes complement known clusters or clusters consisting solely of nBsX genes.

Reviewer #2

In this manuscript, the authors (Santini et al.) sought to understand which mechanisms or pathways control gene imprinting in mouse blastocyst stage. Toward this end, they first identified 71 novel imprinted genes, by RNA-sequencing the hybrid mouse embryos, calling the allelically skewed gene expression, and intersecting with public resources of mouse imprinting genes. Next, they performed the micro-whole-genome bisulfite sequencing (μ WGBS) of individual uniparental blastocysts to examine if the imprinted genes are associated with allelic DNA methylation. Further, they took the published H3K27me3 data to examine whether the DNA methylation-independent imprinted genes are associated with H3K27me3 marks. Finally, they used the maternal Dnmt3l KO or Eed KO datasets to validate their results. The analysis is decent, and the manuscript is written well. However, I have several major and minor concerns.

Major points:

(1) The discovery and methods present in this study are very similar to that in Inoue et al, 2017. What are the main advantages and discoveries in this study? That will be great if the authors can clarify it and make it clearer to the reader.

Thank you for raising these issues, which we would like to clarify. Inoue *et al.* identified a set of 76 H3K27me3-dependent imprinted genes in mouse preimplantation embryos. In our study, we incorporated these published non-canonical imprints as part of the set of 'published imprinted genes'. Thus, all nBiX and nBsX genes lay outside of this set and indeed all other putative imprinted gene sets, and they have not been reported before as imprinted genes.

Inoue *et al.* adopted an approach that was different to ours. They identified parent-of-origin specific expression by intersecting DNaseI-hypersensitivity with androgenote and parthenogenote morula expression data (we used neither approach; see below). Of the 76 genes containing non-canonical imprints reported by Inoue *et al.*, 48 were detectably expressed in our dataset. Filtering for robustly-expressed genes that were not part of the X-chromosome reduced this number to 29 genes. The 28 remaining genes (76 minus 48) were part of the gene definitions used by us, but they did not pass initial expression-threshold filtering and expression was too low for inclusion. We therefore conclude that a large portion of reported non-canonical imprinted genes are expressed at very low levels, if at all, in biparental blastocysts.

Our approach is distinctive because we directly assessed parent-of-origin-specific expression of genes at the blastocyst stage using reciprocal crosses of biparental embryos produced by natural mating (avoiding *in vitro* culture). This made no assumptions about the mechanism(s) underlying parent-of-origin specific expression and allowed us to detect allele-specific expression with a high level of precision, as evidenced (for example) by the high rate of validation by RT-qPCR/Sanger sequencing in independently-derived blastocysts. We utilized parthenogenotes and androgenotes to define DMRs, but direct methylome analysis was not part of the work by Dr. Inoue. One can be confident that our DNA methylation data is highly robust: the set of 859 DMRs genome-wide included 23 of 24 the known germ line (GL)-DMRs.

We now clearly state (Results, I.269) that non-canonical imprinted genes detected by Inoue *et al.* are included in our definition of published imprinted genes. Furthermore, in Figures 4 and 5 we analyse HCon repository and non-canonical imprinted genes as subsets of published imprinted genes. Although most of the nBsX and nBiX genes discovered in our study depend on H3K27me3, we also identify novel imprinted genes that are apparently DNA methylation-dependent. We hope that together, this clearly distinguishes our work from that of others, including Inoue *et al.*, both in the technical approaches we employ, and in that we report distinctive and novel (as well as previously-reported) imprinted genes.

(2) Given that the imprinting status of genes depends on the cell types and developmental stages, I just wonder whether the Dnmt3l KO and Eed KO dataset used in the analysis matches the developmental stages in which the authors characterised gene imprinting.

Thank you and we agree with this point. Where reported, the authors of the cited papers used embryos ~78h after fertilization, which corresponds to the late morula and morula-blastocyst transition stages in wild-type (WT) mouse preimplantation development. However, we are somewhat wary of strict developmental comparisons between embryos with different backgrounds and genotypes, because their developmental rates have never been reported but may differ. For example, it is unclear whether *Eed*-depleted embryos develop at the same rate as WT, and although it is reasonable to infer that the rate is similar, this confounds over-reliance on precise temporal comparisons between them. However, in the context of these previous studies, our analyses show that nBsX and confirmed published imprinted genes exhibit a clear dependence for parent-of-origin specific expression on maternal *Eed* (*mEed*) and, to a lesser extent, *mDnmt3l*.

This analysis is now shown in Figure 5a. In it, we restrict ourselves to nBiX and nBsX genes that clearly exhibit significant monoallelic expression according to criteria explained in the manuscript. We confirmed the imprinting states of multiple genes and correlated them with maternal DMR establishment and H3K27me3 deposition.

Taking the comments of Reviewer #2 on board, we have now completely restructured Figure 5 and sought to clarify the text (l. 305-362). We hope the Reviewer agrees that the data are now presented in a much simpler and statistically stronger manner.

(3) In Line 79, the authors reported that 100 published imprinted genes were absent from the BiX dataset, including *Igf2*, *H13*, and *Commd1*. However, in Line 85, they tried with different statistical methods and only found 24 out of 134 published imprinted genes show significant biallelic expression. What about the rest 76 genes (100 - 24)?

We apply two tiers of statistical analysis to determine equivalently-expressed and parent-of-origin specific genes. Genes with parentally-biased expression were identified based on statistically significant allelic skewing. For the BiX set, this was followed by a further level of stringency to ensure that the genes fulfilled the 70:30 allelic expression ratio.

To identify equivalently-expressed genes (l.83), we tested against the null hypothesis that there actually was a difference (H_0 : absolute $\log_2FC \geq 1$); this is the opposite of 'conventional' differential gene expression tests that aim to reject the null hypothesis that there was *no* difference (H_0 : $\log_2FC = 0$). In this way, only genes with robustly similar allelic expression levels pass the test of statistically significant biallelic expression.

We wish to point out that these two tests do not simply give complementary results. For example, a high *p*-value in the first test means that there is no statistically significant difference between the two alleles. But from this it cannot be inferred that both alleles are expressed to the same levels; replicates may, for example, lack sufficient consistency, in which case both tests (for allele-specific expression or equivalent expression) will give high *p*-values.

(4) Line 235, the authors said "the vast majority of both nBiX and nBsX genes exhibited paternal expression with maternal 5mC at the closest DMR, similar to published imprinted genes". However, in Line 239, they added that "nBiX and nBsX did not show such as association". These two claims seem opposite.

We thank the Reviewer for pointing this out and agree that the statements are confusing. The first statement is based on the correlation between DMRs and parent-of-origin-specifically expressed genes, without any distance cutoffs (we have now included the qualifier 'at any distance' in line 235). Once a cutoff is introduced (e.g. <250kb), this association was no longer detectable. We now explain this explicitly in the text.

Minor questions:

(1) Line 58, the authors excluded the potential X-linked imprinted genes. Can the authors state the reason why X-linked genes are removed from the analysis? Mixed gender embryos? X chromosome inactivation? Or other reasons.

The Reviewer is correct: the reason is that the embryos were of different sexes. We now state this clearly in the methods section. *“Moreover, genes on the X-chromosome (analysed embryos were not matched for sex) and genes with fewer than ten SNP spanning reads in at least one sample were removed from further analysis.”* (l.807).

(2) Line 76, missing the second bracket.

Thank you. Corrected.

(3) In Fig 1, the authors reported that expression of *Commd1* in blastocytes showed mixed pattern, i.e. some are biallelic expression while some are maternal- or paternal specific expression pattern. Can the authors add some details about the original study that *Commd1* is imprinted and which tissues or developmental stage they used, to make the readers clear about the discrepancy.

We thank the Reviewer for this comment, prompted by which we have attempted to clarify this point. *Commd1* (or *Murr1*) is known to be biallelically expressed in embryonic and neonatal mice (Nabetani *et al.*, 1997; DOI: 10.1128/mcb.17.2.789). It was reported to acquire predominantly maternal expression only in mouse adult tissues, particularly in the adult brain. Maternally-restricted expression of *Commd1* in the adult brain is the result of transcriptional interference of the paternally expressed *Zrsr1* gene (maternally methylated), located in the first intron of the *Commd1* gene and transcribed in the opposite direction (Wang *et al.*, 2004; DOI: 10.1128/mcb.24.1.270-279.2004). We now discuss this in the Results section (l.378-379).

(4) In extended Fig 1g, the authors should specify what “KO” stands for. Do these “KO” data here support this paper’s conclusion?

Thank you. The use of “KO” indicates an *Xist* KO cell line. We have now clarified this in the relevant figure, its accompanying legend and in the methods. The *Xist* KO has no apparent impact on the allele specific expression of nBiX and nBsX genes.

(5) Line 210-215, there are 859 DMRs in total, including 410 from clusters C1 and C4, 62 from cluster C5, 349 from cluster C2, and 37 from cluster C5. $410+62+349+37=858<859$

We thank the Reviewer for spotting this typo. DMR-C5 contains 63 and not 62 DMRs. This has now been corrected in the text.

(6) Line 214, the authors said that “most differential DNA methylation is encoded within gamete genomes”. This claim is not appropriate, because cluster C2 and C3 sites are re-established group during preimplantation development that nearly account for 45%.

We agree with the Reviewer that a substantial proportion (~45%) of DMRs are apparently reestablished during development, mostly by selective loss of DNA methylation from one allele. However, this implies that, consistent with our statement, >50% of DMRs are encoded in the germ line. Please note that in the revised version, we have renamed methylation clusters from 1-5 (top to bottom) in the hope that it is easier to follow.

(7) I appreciate the authors tried to provide much information in the Figure 3d, but it’s very difficult to understand and the legend is not clear.

We have now replaced Figures 3d-f (now 2d-f) with a new DMR cluster-based analysis. We have also removed the numbers of known/novel DMRs to avoid overloading the figure. We

hope that Reviewer #2 agrees that these data and the presentation based on the percentage of overlapping regions are now clearer and more intuitive.

(8) Line 217, “overlapped” should be “overlap”.

Thank you. This has been corrected.

(9) Line 335, the authors found that in some cases the differential DNA methylation and parent-of-origin-specific expression is unlinked. I am wondering whether it depends on the location of this DNA methylation in gene body or promoter. For example, gene body DNA methylation doesn't link to the allelic expression and promoter methylation does.

Fig. R9 shows that there is an overall correlation between DMRs and allelically-skewed expression (left). In BsX genes, this relationship is detectable between DMRs on promoters and DMRs up to 100kb either side. Promoter DMRs have a statistically stronger impact than distant DMRs on allele-specific gene expression.

Figure R9. Relationship between the distance of a gene to the next DMR to its allele specific expression state. Statistical significance is indicated by *. n.s. refers to not significant. Promoter-linked DMRs results in a statistically significant in allelically skewed expression.

(10) That will be great if the authors can add the parent-of-origin (maternal or paternal) for imprinting genes to their Figure 6.

Thank you and we agree. We have accordingly completely redesigned Figure 6 and now show genome snapshots including imprinted genes, H3K27me3 peaks and DMR information to provide a more comprehensive overview of imprinted gene clusters.

Reviewer #3

In their manuscript 'Novel imprints in mouse blastocysts are predominantly DNA methylation independent', Santini et al. present a multi-layered analysis of the imprinting landscape in mouse preimplantation embryos. Their study follows previous work demonstrating that some imprints are transient during this stage and do not depend on parent-specific DNA methylation as classical imprints do, but rather rely on maternally-inherited H3K27me3—a phenomenon that has been termed 'non-canonical imprinting' (Inoue et al., Nature 2017). Here, the authors sought to determine the extent to which each of these mechanisms contributes to the regulation of imprinting in the early embryo. By employing allele-specific RNA-seq of reciprocal F1 hybrids, DMR profiling of uniparental and biparental embryos, and analysis of published ChIP-seq and Hi-C datasets, they identified more than 70 putatively imprinted genes, most of which are associated with parent-of-origin-specific H3K27me3 rather than a DMR. This is a carefully executed and comprehensive study that highlights the impact that different regulatory layers may have on imprinting. However, the following issues should be addressed before this manuscript can be considered for publication.

We thank Reviewer #3 for these supportive remarks and now address their specific comments one-by-one.

Major comments

1. Studying the dynamics of imprinting regulation, which is uniquely stable in early embryogenesis despite immense changes to the epigenome, is important due to its dire implications on development. Although the findings of this work argue in favor of a predominant role for H3K27me3 in regulating imprinting during early development, it is still unknown whether the observed parent-specific expression patterns are functionally important or if they merely reflect a byproduct of the numerous differences between the parental genomes at these stages. This is a fundamental point that should be discussed.

We very much agree with Reviewer #3. The question as to why such a large number of genes appear to be specifically imprinted in pre-implantation development is intriguing. Prompted by review, we have now expanded our analysis to show that most parent-of-origin biased gene expression is already encoded by gametic H3K27me3.

The extent to which nBiX and other blastocyst imprinted gene expression is required for development awaits further, and extensive, experimental analysis. However, we have extended our approach by asking in which tissues nBsX and nBiX gene expression is most enriched. This revealed that nBsX and nBiX transcripts share (with those of published confirmed imprinted genes) enrichment in embryonic day (E)14 brain tissue. Enrichment in placental tissues was detected in published imprinted genes, but not nBiX or nBsX. We now describe this finding in I.116-123). We further tested whether the imprinting switch between E3.5 and E6.5 could indicate a functional role in the naïve-to-formative pluripotency transition. To this end, we investigated which BsX genes are dynamically expressed during exit from naïve pluripotency (e.g show differential expression between ESCs cultured in 2i medium and 24h after release from 2i; Lackner et al., 2020, bioRxiv) and found that as a group nBsX genes show more dynamic expression during the exit from naive pluripotency compared to control gene-groups. These data are shown in Supplementary Fig.1k. However, we did not detect a substantial overlap of imprinted genes with hits from screens for factors mediating ESC differentiation, suggesting that BsX genes play no role in the transition from pre- to post-implantation pluripotency (see Fig. R2).

2. Previously described 'non-canonical imprinted genes' are thought to be exclusively paternally-expressed due to maternally-derived H3K27me3. However, some of the genes reported as novel BiX genes (such as Emc2) were found to be maternally-expressed, suggesting paternally-derived H3K27me3. This point should be studied further.

We identify a set of maternally expressed genes in our dataset, and eight were indeed marked by H3K27me3 in sperm and therefore inherited through the germ line. We now show this in Figure 4, which has been added to the revised manuscript.

3. A crucial aspect of identifying novel imprinted genes is demonstrating not only parental-allele-biased expression but also parent-of-origin epigenetic inheritance. Therefore, the authors should analyze H3K27me3 in oocytes and sperm to determine whether the parent-specific patterns observed in embryos are inherited from the gametes (complementing the analysis of DMRs in Fig 3c).

We completely agree and thank the reviewer for this suggestion. We have accordingly now dedicated an entire new figure (Fig. 4) to this issue. We found that, indeed, the majority of paternally-expressed genes inherit H3K27me3 through the (mainly female) germline. We also detected a correlation between paternally-inherited H3K27me3 and maternally expressed genes (see preceding point).

4. The authors should elaborate on how many of the 40% of novel BiX/BsX genes that depend on either maternal DMRs or H3K27me3 based on KO morulae (Fig. 5e), depend on each or both mechanisms. These data are directly linked to the overall aim and conclusion of the paper.

We have sought to improve our analysis and presentation of data presented in Figure 5. We have optimized our statistical approach to determine dependence on *mEed* or *mDnmt3l* activity and now clearly identify which genes depend on either or both mechanisms for imprinted gene expression. We now address the Reviewer's question in Figures 5a, b, f and g (l.305-362). See also our response to point 7, below.

5. How many of the detected BiX/BsX genes were already identified by previous studies reporting non-canonical imprinting? It is important to discuss these numbers as an indication for reproducibility, as well as for the feasibility to identify new genes regulated by this putative mechanism.

None of the nBiX or nBsX genes were reported previously as non-canonical imprinted genes; We included all published non-canonical imprints (that is, including the ones reported by Inoue *et al.*) in the list of published imprinted genes. We further clarify this in the main text (l.269-270) and the Methods section.

Of the 76 genes exhibiting non-canonical imprints reported by Inoue *et al.*, 29 were present with sufficient read count over SNPs (after excluding X linked genes) to allow meaningful conclusions. Twenty of these were part of the pubBsX (published blastocyst skewed expressed) genes and therefore validated in our analysis.

6. The monoallelic expression of several of the novel BiX/BsX genes were confirmed by Sanger sequencing. How many of these novel genes were tested in total, and what was the false discovery rate?

Thank you for pointing out this omission. We now include the missing information in the main text and have updated Supplementary Table 3. Overall, Sanger sequencing validated allelic bias in expression of 10 of 11 tested nBsX and nBiX genes (all of the genes for which we could obtain useable data) in independent blastocyst samples, and similarly, 8 of 8 published imprinted genes.

7. The data in Fig. 5a–c should be accompanied by a statistical analysis to help determine the extent of similarity between the observed molecular phenotypes in control vs. KO embryos. It would also be helpful to explain this complex Figure in more detail in the text and/or in the Figure legend.

We agree and have spent time improving our analysis and working out how to present it clearly without losing information in Figure 5. We have accordingly now re-analyzed published datasets of *mDnmt3l* and *mEed* KO and applied more stringent statistical cutoffs to define first, imprinted gene expression in WT embryos, and then to identify which imprinted gene expression changes significantly in KO embryos (see also Fig. R10)

Figure R10 (see also manuscript Figure 5). Functional dependence of novel candidate genes on maternal H3K27me3 or maternal DNA methylation. **a** Heatmap indicating allelic expression bias of BsX genes in wt morulae or morulae carrying maternal genetic deletions of either *Dnmt3l* (*mDnmt3l* KO) or *Eed* (*mEed* KO). Colours distinguish between pubBsX (with further indication for genes belonging to the high confidence (HCon) repository imprints or the published non-canonical (H3K27me3-marked) imprint category (grey/black squares), nBsX and nBsX other than BiX genes. Only genes with significant allelic bias (adj. $p < 0.1$) in at least one WT morula were included in the analysis. *, adj. $p < 0.1$; **, adj. $p < 0.01$; ***, adj. $p < 0.001$. Allelic expression bias is shown in the first two columns of each WT-mKO set (colour coded from red to blue). The third column of each WT-mKO pair indicates mKO induced changes in the allelic expression bias (colour coded from red to blue; *, adj. $p < 0.05$; **, adj. $p < 0.01$; ***, adj. $p < 0.001$); only genes that showed a reduction in their allelic bias upon maternal *Dnmt3l* or *Eed* deletion were considered) **b** and **c** Pie charts indicating gene numbers within respective groups (pubBsX and nBsX) losing parent-of-origin specific expression following maternal deletion of either *Dnmt3l* (dependent on *mDnmt3l*), *Eed* (dependent on *mEed*) or both (dependent on both) in morulae. Genes that not dependent on either are also indicated. **d** and **e** Box plots illustrating how allelic ratio (absolute log2FC) of pubBsX or nBsX genes is affected by maternal deletion of *Dnmt3l* (*mDnmt3l* KO) or *Eed*

(*mEed* KO) at the morula stage. Only genes with significant allelic bias (adj. $p < 0.1$) in at least one WT morula were included. Wilcoxon signed-rank tests were performed for WT vs KO comparisons (WT-1 vs *mDnmt3l* KO and WT-2 vs *mEed* KO), and for comparing the WT vs KO differences between datasets. *, $p < 0.05$; **, $p < 0.01$; ***, $p < 0.001$; ns, not significant. **f** and **g** Bar charts indicating associations between functional response to loss of either *mDnmt3l* or *mEed* (as defined in Fig. 5b) with physical proximity to DMRs (within 250 kb or in the same TAD) or the presence of TSS-associated (± 5 kb) H3K27me3 for *pubBsX* and *nBsX* genes.

8. The conclusions drawn from Fig. 1h (starting from page 7, last paragraph) are not supported by statistically significant results.

We agree and apologize for previously including these data, which are uninformative. They have now been replaced. For a more informative portrayal of the role of (n)BiX genes during development, we instead determined whether BsX genes are specifically enriched in certain tissues based on ENCODE expression data. This analysis revealed that nBiX gene expression is enriched in E14 brain. These data are now presented in Figure 1g.

Minor comments

1. The claim made in the Abstract that “neither parent-of-origin-specific transcription nor DMRs have been comprehensively mapped” is untrue. There is a multitude of papers that have done so and should be cited (to name a few: Court et al., *Genome Res.* 2014; Babak et al., *Nat. Genet.* 2015).

This is of course entirely correct, and we apologise for over-reaching. The statement has accordingly been toned down to reflect that we are referring to a particular developmental stage, and the passage now reads: "However, neither parent-of-origin-specific transcription nor imprints have been comprehensively mapped at the blastocyst stage of preimplantation development."

2. The term ‘blastocyst-imprinted expressed’ (BiX) genes was defined based on parent-of-origin allele-specific expression alone. However, to qualify as truly imprinted, parent-specific epigenetic marks need to be identified. Therefore, this set of genes should be regarded as ‘potentially’ or ‘putatively’ imprinted.

The term BiX refers to parent-of-origin specific expression, based exclusively on expression data without any prior information on potential regulatory mechanisms. We show that most nBiX and nBsX genes are independent of DMRs but largely rely on H3K27me3, which, in the revised manuscript, we now show to be inherited through the germ line and largely dependent on maternal *Eed* activity (Figs 4 and 5). Therefore, with the exception of four nBsX genes, all nBsX genes could be associated with either differential DNA methylation or allele-specific H3K27me3 (see manuscript Figs 4 and 5).

3. The statement that somatic DMRs “carry marks at the blastocyst stage that are not directly dependent on DNA-methylation” (page 9, starting from line 155) is not entirely accurate. For example, the *Meg3* somatic DMR depends on differential methylation of the IG-DMR at this locus.

We have analysed this in greater detail and found that the *Nesp*, *Cdkn1c*, *Meg3* and *Igf2r* promoter-associated somatic DMRs were differentially methylated within a 250kb window in blastocysts and it is indeed possible that these distal DMRs serve to seed methylation of their associated alleles later in development. Although somatic DMR acquisition may be guided by neighbouring DMRs during post-implantation development, the mechanism of this acquisition is unknown. We discuss this in the results section (l. 153-156).

4. In “thirty-seven blastocyst DMRs exhibited little or no DNA methylation in oocytes or sperm” (line 212), the percentage out of all DMRs should be indicated to highlight this cluster of genes represents a minority of DMRs.

Thank you for pointing this out. We have changed the text to: " A minority of blastocyst DMRs (37; 4%) exhibited little or no DNA methylation in oocytes or sperm (DMR cluster C1)...".

5. The intriguing finding that "confirmed published imprinted genes showed only limited response to loss of *mDnmt3l*" (line 285) should be discussed.

We have restructured analysis, shown in Fig. 5, to improve the statistical strength of the conclusions. We find that, overall, most published confirmed imprinted genes (now termed pubBsX genes) are dependent on *mEed*. Even after separating non-canonical imprinted genes and analyzing exclusively imprinted genes which have been reported in multiple imprinting repositories (and which are closely associated with DMRs according to our data; they are termed HCon genes in our manuscript), 4 out of 9 show some dependence on *mEed*. However, in contrast to all other analysed gene groups, for the group of HCon genes the majority of genes (5 out of 9) showed dependence or co-dependence on *mDnmt3l*. This is described in the manuscript in l. 316-326 and in Figures 5d, e and Suppl. Fig. 5 b, c) and referred to in the discussion section. (l.420-430).

6. In "expression of some 50% of novel imprinted genes were apparently H3K27me3- and DMR- independent" (line 303), it should be clarified that this relates only to maternal H3K27me3 and DMRs.

Thank you and we agree. Accordingly, this is now clearly stated ("*The effect of maternal Eed depletion was exclusively detected in paternally-expressed genes.*"; l.325-326).

7. Figure improvements:

– What are the WT/KO genotypes in Extended Fig. 1g?

We apologise for this omission and have now added the missing information in the Figure, legend and methods. KO refers to an *Xist* knock out, which has no detected impact on nBsX or nBiX gene expression.

– I found Fig. 3d–f challenging to interpret. Also, it is unclear what the '+' in parentheses represent

Thank you; on reflection, we agree. The analysis presented in Figure 3d-f (new Fig. 2d-f) has accordingly been refined so that it is clearer. This includes presentation of the percentage overlap between specific DMR clusters and defined genomic features or ChIP and motif enrichment analyses. We hope Reviewer #3 agrees that these new representations are clearer and more intuitive.

– Fig. 5a–c are very important but hard to grasp. Consider simplifying them

To improve clarity, which we agree is important, we have now refined our analysis and increased the statistical strength of our conclusions. Key information in Figure 5a is now presented as a heatmap, which we hope is more accessible.

– Fig. 7 could be improved by adding close-up views of representative clusters.

Thank you. Prompted by this helpful comment, we now show data from Figure 6a and Supplementary Figure 6 (imprinting clusters) as a set of close-up views intended to encapsulate all relevant information; this is really a synopsis that we hope will be a useful resource. Figure 6b provides an overview of cluster data, BsX genes and DMRs. We believe that these changes improve accessibility of our data and provide a visual overview of the genome structure pertaining to published imprinted genes and the novel ones reported here.

Reviewer Comments

Reviewer #1 (Remarks to the Author):

The authors have clearly answered my different concerns. The additional analysis, including the TAD and H3K23me3 domains, have been performed.
I support the manuscript for publication in Nature Communications.

Reviewer #2 (Remarks to the Author):

The authors have addressed the concerns that were raised. Now this revised manuscript has been improved considerably.

Reviewer #3 (Remarks to the Author):

In their revised version of the manuscript, the authors have provided adequate responses to my concerns. I recommend publication without further revision.